# The Hippo Pathway Effectors YAP/TAZ-TEAD Oncoproteins as Emerging Therapeutic Targets in the Tumor Microenvironment

**DOI:** 10.3390/cancers15133468

**Published:** 2023-07-02

**Authors:** Reza Bayat Mokhtari, Neda Ashayeri, Leili Baghaie, Manpreet Sambi, Kosar Satari, Narges Baluch, Dmitriy A. Bosykh, Myron R. Szewczuk, Sayan Chakraborty

**Affiliations:** 1Department of Pharmacology and Therapeutics, Roswell Park Comprehensive Cancer Center, Buffalo, NY 14263, USA; reza.bayatmokhtari@roswellpark.org (R.B.M.); dmitriy.bosykh@roswellpark.org (D.A.B.); 2Department of Biomedical and Molecular Sciences, Queen’s University, Kingston, ON K7L 3N6, Canada; 16lbn1@queensu.ca (L.B.); m.sambi@queensu.ca (M.S.); szewczuk@queensu.ca (M.R.S.); 3Division of Hematology and Oncology, Department of Pediatrics, Ali-Asghar Children Hospital, Iran University of Medical Science, Tehran 1449614535, Iran; neda.ashayer@gmail.com (N.A.); sattarikosar99@gmail.com (K.S.); 4Department of Immunology and Allergy, The Hospital for Sick Children, Toronto, ON M5G 0A4, Canada; narges.baluch@sickkids.ca

**Keywords:** YAP/TAZ, TEAD, Hippo signaling pathway, carcinogenesis, tumor microenvironment, cancer, drug resistance, immunotherapy, gut microbiota, combination therapies

## Abstract

**Simple Summary:**

YAP/TAZ are the central effectors of the Hippo pathway and orchestrate their oncogenic program by binding to TEAD transcriptional factors. Here, we document a comprehensive understanding of how YAP/TAZ dependent tumors could be exploited for improving cancer therapies. Aside from providing the recent updates to the YAP/TAZ oncogenic pathway, we develop an exciting perspective on developing inhibitors that intervene with YAP/TAZ signaling with a close focus on the immune-microenvironmental regulations.

**Abstract:**

Various cancer cell-associated intrinsic and extrinsic inputs act on YAP/TAZ proteins to mediate the hyperactivation of the TEAD transcription factor-based transcriptome. This YAP/TAZ-TEAD activity can override the growth-limiting Hippo tumor-suppressor pathway that maintains normal tissue homeostasis. Herein, we provide an integrated summary of the contrasting roles of YAP/TAZ during normal tissue homeostasis versus tumor initiation and progression. In addition to upstream factors that regulate YAP/TAZ in the TME, critical insights on the emerging functions of YAP/TAZ in immune suppression and abnormal vasculature development during tumorigenesis are illustrated. Lastly, we discuss the current methods that intervene with the YAP/TAZ-TEAD oncogenic signaling pathway and the emerging applications of combination therapies, gut microbiota, and epigenetic plasticity that could potentiate the efficacy of chemo/immunotherapy as improved cancer therapeutic strategies.

## 1. Introduction

The Hippo pathway, named after its main component, the protein kinase Hippo (Hpo), is a highly conserved pathway involved in cellular proliferation, apoptosis, and differentiation. It represents one of the more recent organ size-limiting pathways discovered [1,2,3,4]. YAP (Yes-associated protein) and TAZ (transcriptional co-activator with PDZ-binding motif) are components of a more prominent pathway called the “Salvador-Warts-Hippo” (SWH) pathway [5]. The YAP/TAZ complex is fundamental to maintaining many cellular processes, tissue homeostasis, and proper organ development. However, these proteins’ role in cancer cells facilitates tumor progression by altering their downstream signaling effectors. As a result, YAP and TAZ present an excellent opportunity to develop therapies targeting this critical pathway.

This review builds on the current body of literature by providing insights on the role of YAP/TAZ as effectors of the Hippo pathway in healthy versus cancer microenvironments. Importantly, we provide an overview of emerging advancements in the interplay between the Hippo pathway, the immune system, and immunotherapy within the tumor microenvironment, with YAP/TAZ as central players. Additionally, our review highlights the emerging relationship between the Hippo pathway, the gut microbiota, and epigenetic regulation. Taking these multi-disciplinary roles into consideration, we discuss attractive strategies that may potentially enhance the effectiveness of immunotherapy. Taken together, the integration of recent research and our group’s work in this domain adds valuable perspectives that illustrate the complex interplay between the Hippo pathway, the immune surveillance mechanisms, and the gut microbiota, reinvigorating the need for further investigation in this area.

## 2. An Overview of the Drosophila Core Hippo Pathway

The Drosophila core Hippo pathway is a signaling cascade that plays a critical role in regulating tissue growth and organ size in fruit flies. The core components of SWH include the Hippo kinase bound to the adaptor protein Salvador, which activates the LATS/Warts kinase [5]. The role of this pathway was not well understood until it was studied in *Drosophila* tissues. In a landmark study, Yorkie (the YAP/TAZ ortholog in Drosophila) was used to experiment on the inactivation of the Salvador–Warts–Hippo pathway, which saw an increase in cell proliferation and the emergence of tumors [6]. Following the discovery of an oncogenic role for YAP, its paralogue, TAZ, was eventually shown to promote invasiveness and tumorigenic potential [7]. The significant milestones, based on the discovery of the core and Hippo effectors, that contributed to the understanding and functional roles of critical Hippo components are presented in Figure 1.

## 3. The Core Hippo Kinase Cassette in Mammals

The kinase cascade of the Hippo pathway can be divided into two main branches: (a) the serine/threonine kinase core sterile 20-like kinase ½ (MST1/2) and (b) large tumor suppressor 1/2 (LATS1/2). MST1/2 kinases can activate LATS1/2 kinases and MOB1A/B (as adaptors) by binding the regulatory protein SAV1/WW45 [33]. More recently, Tao kinase and MAP4K have been shown to directly phosphorylate LATS1/2, thus acting parallel to ST1/2 [28,34]. The activated LATS1/2-MOB1A/B complex phosphorylates YAP and TAZ to activate the Hippo cascade [1,15,35,36]. The phosphorylation of YAP and TAZ also leads to the sequestering of the complexes in the cytoplasm, thus inhibiting the activation of their target genes [37] (Figure 2). Previous studies have noted that a mutation in the serine residues of YAP and TAZ results in decreased sensitivity to inhibition of the complex by the Hippo pathway [38]; therefore, these may be of interest in developing therapeutics.

### Upstream Regulators of the Hippo Cascade in Mammals

Multiple extracellular cues impact the Hippo cascade upstream of the Hippo kinases. An upstream component of the Hippo cascade is Merlin, a protein encoded by NF2 that is essential for maintaining tight cell-cell junctions [39]. When the NF2 tumor suppressor locus is mutated, it often results in neuronal tumors [40]. Regarding its function in YAP/TAZ activity, Merlin/NF2 is thought to allow the activation of LATS by directly binding to it, which, as previously discussed, is necessary to phosphorylate YAP [41,42,43] (Figure 2). Merlin is shown to exist in open and closed forms that are associated with its tumor-suppressor abilities [44]. PAK1-mediated phosphorylation of Merlin at Ser518 inactivated Hippo activation, suppressing its anti-tumorigenic functions, an effect that AMOT proteins oppose [44]. Though this phosphorylation of the Ser518 residue does not alter Merlin’s conformation but instead prevents angiomotin from binding and thus inhibits the core Hippo pathway activation. Furthermore, the AMOT family of proteins, by binding to LATS1 and SAV1-MST1, acts as a scaffold that connects YAP to LATS1 [45]. Scribble, required for maintaining cell polarity, is another regulator of the Hippo cascade as it aids in assembling MST, LATS, and TAZ [46,47]. It has also been linked to triggering epithelial-to-mesenchymal transition (EMT), allowing tumor progression [48]. EMT also activates YAP and TAZ, which further assist the tumor in acquiring traits that allow it to survive and progress [46]. Mechanical stress signals such as tension and compression also promote the recruitment of F-actin-binding proteins, which then trigger the activation of Hippo/MST1/2 kinases [49]. As shown in Figure 2, another upstream input of the Hippo cascade is G-protein-coupled receptors (GPCRs), which can activate the Hippo pathway by binding to ligands such as LPA (lysophosphatidic acid) or S1P (sphingosine-1-phosphate) [25]. Activation of GPCRs leads to the activation of Gα12/13, which in turn activates Rho GTPases such as RhoA and Rho kinase (ROCK). Active RhoA promotes the phosphorylation and inactivation of the downstream Hippo pathway effector, YAP (Yes-associated protein), leading to its cytoplasmic retention and inhibition of downstream target gene expression.

Cell–cell adhesion molecules, such as E-cadherin and α-catenin, have been implicated in the regulation of the Hippo pathway. Disruption of the E-cadherin and α-catenin complexes has also been shown to interfere with the phosphorylation of YAP, decreasing it, which in turn increases the amount of YAP found in the nucleus [35]. These molecules contribute to the establishment of cell polarity and the formation of adherens junctions. Adherens junctions, in turn, regulate the Hippo pathway by influencing the localization and activity of components such as Merlin (also known as NF2), which acts upstream of MST1/2 and LATS1/2 to inhibit YAP activity. In addition, members of the membrane-associated guanylate kinase (MAGUK) protein disc large homolog DLG family connect cell polarity and Hippo signaling through PAR-1 linkage with MST1/2 [50]. While the phenotypic states of cancer cells are largely dynamic, they often switch between the classical ‘r’ states that are proliferative and responsive to cell density and the ‘K selection’ states that have limited growth potential [15]. Interestingly, Li et al. found that the YAP phosphorylation inhibitor DLG2 differentially impacts the YAP/TAZ transcriptome between two distinct environment-adapted cell lines and is associated with cell survival [51]. The study found that the presence of the YAP phosphorylation inhibitor DLG2 was associated with cell survival in differentially expressed genes between two environment-adapted cell lines. DLG2’s inhibitory effect on YAP phosphorylation protected cells and contributed to their adaptive response to diverse environments. These findings highlight DLG2’s role as an upstream regulatory factor in the Hippo pathway, influencing tumor behavior and functional diversity.

Other kinases have also been found to activate components of the Hippo pathway. The striatin-interacting phosphatase and kinase (STRIPAK) complex is an example of one of these kinases. The complex contains the phosphatase protein phosphatase 2A (PP2A), which dephosphorylates MST1/2, activating YAP/TAZ [52]. Thousand-and-one amino acid kinases (TAO) also activate the MST1/2 kinases; however, this activation is mediated through the direct phosphorylation of MST1/2, in contrast to the STRIPAK complex [53].

**Figure 2 cancers-15-03468-f002:**
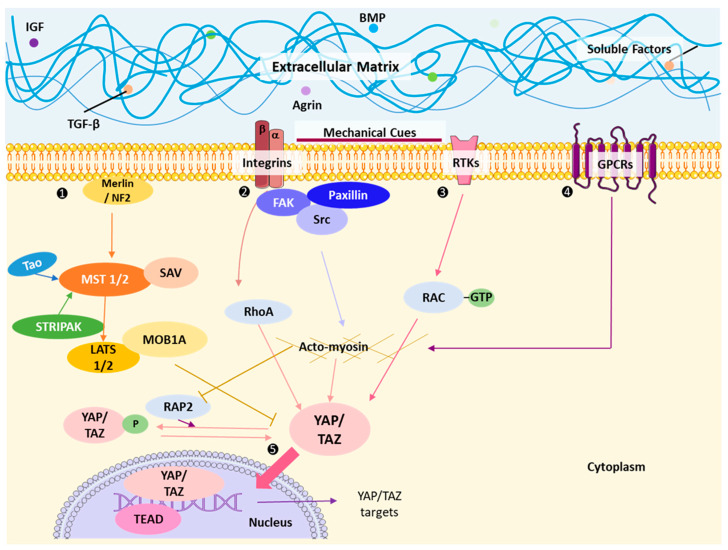
Overview of YAP/TAZ regulation by Hippo and cell-ECM mechanotransduction. Multiple extracellular cues impact the Hippo cascade upstream of the Hippo kinases. (1) The core of the Hippo pathway consists of a kinase cascade composed of the serine/threonine kinases MST1/2 and LATS1/2, which phosphorylate and activate MST1/2 and require MOB1A/B. MST1/2 then associates with and activates LATS1/2 by phosphorylation, subsequently phosphorylating and inhibiting the downstream effectors YAP and TAZ. (2) Mechanotransduction is one of the ways to regulate YAP/TAZ, involving contractile forces from cell-cell interactions, rigidity signals from cell-ECM interactions, and topographical signals for cell/tissue alignment at both tissue and ECM levels. (3) Soluble growth factors and RTK signaling can inactivate the Hippo pathway or independently activate YAP and TAZ via Rho/Rac GTPase pathway, ultimately producing proliferative potential. (4) GPCRs are one of the pathways that repress Hippo activation and thus increase YAP/TAZ signaling. (5) Collectively, these signaling pathways culminate in the YAP/TAZ complex being translocated to the nucleus and binding to TEAD1-4 to activate the transcription of target genes [15,17,25,35,39,40,41,43,44,45,46,47,48,49,52,53,54]. Abbreviations: bone morphogenic protein, BMP; focal adhesion kinases, FAK; G-protein coupled receptors, GPCRs; guanine triphosphate, GTP; insulin growth factor, IGF; large tumor suppressor 1/2, LATS1/2; MOB Kinase Activator 1A, MOB1A; phosphate, P; proto-oncogene tyrosine-protein kinase, Src; Ras homolog family member A, RAP2; Ras related protein 2A; receptor tyrosine kinases, RTKs; Salvador, Sav; striatin (STRN)-interacting phosphatase and kinase, STRIPAK; thousand-and-one amino acid kinases, TAO; transcriptional enhanced associate domain, TEAD; transcriptional co-activator with PDZ-binding motif, transforming growth factor beta, TGF-β; WW-domain-containing transcription regulator 1, WWTR1 (also known as TAZ); Yes-associated protein, YAP.

## 4. Impact of Extrinsic Stimuli and Mechanotransduction on YAP/TAZ

Moreover, several soluble growth factors, such as transforming growth factor-beta (TGF-β), insulin-like growth factor (IGF), bone morphogenetic protein (BMP), epidermal growth factor (EGF), transforming growth factor-alpha (TGF-α), and hepatocyte growth factor (HGF), can regulate the Hippo pathway [17,54]. These growth factors activate receptor tyrosine kinases (RTKs), which initiate downstream signaling cascades. Activation of RTKs can trigger the PI3K/Akt pathway, resulting in the phosphorylation and inhibition of the Hippo pathway components, such as MST1/2 (mammalian Ste20-like kinases 1/2) and LATS1/2 (large tumor suppressor kinases 1/2), ultimately leading to YAP activation and target gene expression. The abundance of these upstream regulators demonstrates multiple ways the Hippo cascade can be dysregulated to promote tumor growth and metastasis. Understanding the role of each member of the Hippo cascade represents a great opportunity as a potential therapeutic target for cancer therapy. Agrin represents another important soluble growth factor implicated in regulating the Hippo pathway; however, additional research is needed.

Mechanotransduction is a process by which biochemical signals are created after a mechanical stimulus has been achieved in the microenvironment [20]. Controlling the Hippo pathway by mechanical upstream signals is highly dependent on the effects of mechanical signals and tension on the cytoskeleton, the tension on cell-cell and cell-matrix adhesions, cell shape, and cell density (discussed further below) [54]. These signals can aid tumor growth and differentiate cells into a metastatic phenotype [20,55]. YAP/TAZ is overarchingly regulated by mechanotransduction at tissue and supra-tissue levels in the extracellular matrix (ECM). Mechanical signals in cells or tissues are generated by the resulting contractile force from cell-cell interactions, rigidity signals between cell-ECM interactions, and the topographical signals by which the cells/tissues align themselves [56]. YAP/TAZ is responsible for the mechanosensory characteristics of tumors and mechanotransduction for dictating the location of the complex in the cells [20,33,57]. The YAP/TAZ complex directly interacts with components of the actin cytoskeleton, such as actin filaments, microtubules, and intermediate filaments, which are critical for signaling through mechanotransduction [35]. More recent investigations have shed insights on the cooperative roles of bulk extrinsic stiffness and overexpression of oncogenes such as RAS or other receptor tyrosine kinases (RTKs) to aggravate YAP/TAZ mechanotransduction and cytoskeletal dynamics to convert normal cells into tumor precursors [58,59]. Independent of the Hippo core cascade, cell stretching, geometrical alterations, and high mechanical forces are sensed by actin-processing factors that regulate YAP/TAZ [49]. These characteristics are demonstrated in breast cancer, where YAP/TAZ is linked to increased ECM rigidity, promoting angiogenesis and invasion of tumor cells [60]. It has been theorized that the mechanical signals that make critical decisions for the cell regarding differentiation and proliferation can activate YAP/TAZ and trigger its proliferation and nuclear localization [20]. When these mechanotransduction signals change the ECM into a morphology lacking the adhesiveness commonly seen, YAP/TAZ gain their proliferative properties and become cytosolic [61].

In contrast, a rigid ECM can maintain YAP/TAZ in the nucleus [20]. Moreover, when the cell geometry is manipulated to make cells round and compact, YAP/TAZ are no longer sequestered in the nucleus, and cells no longer proliferate or differentiate [20,62]. The stiffness of the ECM also differentiates mesenchymal stem cells (MSC) into different cell types. In highly rigid ECMs due to elevated YAP/TAZ, bone differentiation is noticed, while in a more flexible environment, MSCs differentiate into adipocytes [20]. Tumors, often manifesting a rigid ECM, are supposed to harbor a ‘molecular sink’ for mechanotransducing YAP/TAZ. One fine example of this is represented by the proteoglycan Agrin, which is associated with stiff ECM and focal adhesions of liver cancer cells and conveys mechanical signals to YAP/TAZ by shutting the Hippo cascade ‘off’ [63,64,65]. The integrins connect the ECM to cells and are the central mechanosensing receptors [66]. Together with the integrins, the focal adhesion kinases (FAK), Integrin-linked kinases (ILK), and proto-oncogene tyrosine-protein kinase (Src) also sense the ECM mechanical changes, both of which have been shown to activate YAP/TAZ [67,68,69,70]. Furthermore, cell adhesion to the stiff matrix that engages fibronectin-mediated integrin activation was also sufficient to activate YAP/TAZ by inhibiting LATS1/2 activity through a FAK-SRC-PI3K-PDK1 pathway [71]. YAP may sustain this mechanotransduction loop further by promoting focal adhesion assembly and FA-targets [72]. Intracellular mechanotransduction is also mediated by RAP2, a Ras-related protein, which is activated by low stiffness by phosphatidic acid (PA) and inactivates YAP under low stiffness conditions [73]. Mechanistically, the YAP inhibition under compliant ECM was achieved through MAP4K4/6/7 and ARHGAP29.

### 4.1. Regulation by Cell Density and Epithelial Architecture

Cell density and epithelial architecture are vital in regulating YAP and TAZ. In vitro studies have shown that if YAP is unphosphorylated, it can inactivate contact inhibition proliferation (CIP), a fundamental process in cancer progression [15]. This process prevents cell overgrowth during culturing; therefore, a loss of this function is a property of cancer. There have been some proposed mechanisms for the role of CIP in YAP/TAZ phosphorylation, but one of the most commonly cited is E-cadherin and α-catenin’s effect on LATS1/2 [49]. These cell surface proteins have been shown to activate LATS, which phosphorylates YAP/TAZ; cell-cell contact deteriorates YAP/TAZ activity [49]. When cell density is low, there is a reduction in cell-cell contacts, thereby decreasing the activation of YAP/TAZ and increasing the complex’s nuclear localization [54]. In contrast, during high cell density, more cell–cell contacts increase the phosphorylation of YAP/TAZ in the cytoplasm [54]. Moreover, the polarization of epithelial cells is vital for regulating the Hippo pathway, as the dysregulation of the basolateral domain adhering junctions changes the cell’s shape and polarity, triggering YAP/TAZ activity [47].

Annexin A2 (ANXA2) is another transmitter of cellular density working centrally in the Hippo pathway [74,75]. When cell density increases, this protein associates with YAP, shuttling it to the membrane, where it becomes phosphorylated and transcriptionally inactivated [75]. A recent study found that a small molecule named PY-60 frees ANXA2 from the membrane, which frees and activates YAP [75].

In another recent study, KIRREL1, a cell adhesion protein previously only associated with the kidney filtration barrier, has been shown to mediate Hippo signaling. KIRREL1 can sense cell–cell interactions and interact with SAV1 to recruit the molecule to cell-cell contact sites [76]. In the past few years, SAV1 has been shown to work in parallel to MST1/2 as an upstream positive regulator of the Hippo pathway in mammals, a finding that challenged the notion of Hippo as a linear pathway [76,77]. The finding that KIRREL1 interacts with a Hippo pathway component led to further investigations into its exact role. Some of these new findings include how KIRREL1 is broadly expressed in many tissues and how knockouts of the protein lead to increases in the activity of YAP in surrounding cells [76]. Moreover, since KIRREL1 regulates cell-cell adhesion through its extracellular domain, it is hypothesized that cell–cell interactions may regulate KIRREL1-SAV1 via a feedback loop [76]. These new findings demonstrate opportunities to target ANXA2 and KIRREL1 as potential therapeutic options in YAP/TAZ-active cancers.

### 4.2. Regulation by Rho-GTPases and Actin Modulation Machinery

Rho-GTPases (RhoA, Rac1, and Cdc42) regulate the Hippo pathway through modifications of the actin cytoskeletal dynamic and cell adhesion through the phosphorylation of YAP. RhoA activates the Rho-associated protein kinase (ROCK) pathway, sequestering YAP/TAZ to the nucleus through the inhibition of LATS1/2 [25,35]. Rac1 and Cdc42 can regulate the Hippo pathway by increasing cellular tension and disrupting cell–cell junctions, which results in the localization of YAP/TAZ to the nucleus [35]. Statins are also vital to inhibiting the Rho-GTPase pathway as they downregulate YAP/TAZ, and Rho inhibition is linked to YAP/TAZ phosphorylation [25]. In addition to Rho-GTPases, G protein-coupled receptors (GPCRs) are one of the pathways that repress Hippo activation and thus increase YAP/TAZ signaling by reducing YAP-S127 phosphorylation [25]. While G12/13-, Gq/11-, and Gi/o-coupled receptors and agonists activate YAP/TAZ, epinephrine and glucagon were shown to inhibit YAP/TAZ via engaging Gs-coupled receptors [25]. The presence of active G_q/11_ with a Q209L mutation can lead to the dephosphorylation of YAP/TAZ, demonstrating that YAP and G_q/11_ engage in crosstalk [78,79].

### 4.3. YAP/TAZ Nucleocytoplasmic Shuttling

In normal cells, the YAP/TAZ complex is predominantly found in the cytoplasm, where it can be phosphorylated by LATS1/2 [37]. When the complex is dephosphorylated, usually due to growth factor-related signaling or mechanical stress, it is localized in the nucleus [80]. This nucleocytoplasmic shuttling is regulated by importins, which bind to nuclear localization signals (NLS) on the complex and allow them to be transported into the nucleus through pore complexes [81]. In contrast, exportins bind nuclear export signals (NES) to aid transport back into the cytoplasm [81]. They are considered co-transcriptional co-activators that can send signals that help regulate cell proliferation, cancer cell growth, and apoptosis [82]. Their subcellular localization depends on the conditions experienced by the cells/tissues.

The consensus HXRXXS motif in YAP is phosphorylated by LATS1/2 in response to upstream inputs and is conserved at all five residues, including sites (S61, S109, S127, S164, and S381) [83], corroborating the widespread observations that YAP5SA is strictly nuclear and overrides upstream regulation, resulting in increased tumorigenesis. When YAP is mutated at the LATS 1/2 phosphorylation site (S127), inhibition of PDGF receptors causes YAP to move from the nucleus (active) to the cytosol (inactive). During the initial decade of Hippo pathway research, Serine 127 (S89 in TAZ) that affects nucleo–cytoplasmic localization and Serine 381 (that affects YAP degradation by sequential recognition by CK1 kinase and the E3 ligase SCFβ-TRCP) were identified as key phosphorylation sites that suppress the oncogenic functions associated with YAP [83]. Since then, several additional serine and tyrosine phosphorylation sites have been identified that affect YAP function. For instance, Nemo-like kinases (NLK) have been shown to phosphorylate YAP at Ser128, which blocks the interaction with 14-3-3 and increases its nuclear localization [84]. These emerging data are indicative of mutual exclusivity amongst Ser127 and Ser128 residues of YAP. Tyrosine phosphorylation, specifically by SRC family kinases (SFK) targeting YAP at Y357, drives this shuttling [85]. The primary function of SFK-mediated phosphorylation at Y357 is to retain YAP in the nucleus [44]. Dasatinib inhibits SFK, causing YAP to migrate to the cytosol and decreasing YAP target genes. LCK, an SFK member, plays a key role in regulating YAPY357 phosphorylation, according to siRNA knockdown experiments, suggesting a novel function for LCK. This tyrosine phosphorylation method differs from serine phosphorylation, implying separate signals for YAP localization, and is also consistent with the notion that dasatinib-mediated inhibition of SFK impaired tumor growth and metastasis [86]. In addition, c-ABL (Abelson murine leukemia viral oncogene) also phosphorylated YAP at tyrosine residue 357 to suppress its oncogenic functions and stabilize YAP-p73 interactions that induced cell apoptosis [87]. Consistent with these observations, Src has also been shown to inhibit LATS by tyrosine phosphorylation, which exerts enhanced YAP activity and is also linked to intestinal inflammation [88,89].

### 4.4. Nuclear YAP/TAZ Complexes and Transcriptional Output

YAP/TAZ do not have any DNA binding capabilities and therefore require additional DNA-binding partners to associate with chromatin. In this regard, the transcriptionally enhanced associate domains (TEADs) are fundamental for the interaction of transcription factors for YAP/TAZ, preventing their overgrowth [23]. When the YAP/TAZ complex is translocated in the nucleus, it binds to TEAD1-4 to activate the transcription of target genes such as CTGF, CYR61, ANKRD1, and CCND1 [35]. YAP/TAZ prefers to bind to TEAD as a complex, allowing cells to proliferate [80]. Many other transcriptional factors have been shown to interact with YAP/TAZ (AP-1, TRPS1, ZEB1, etc.); however, the role of TEAD emerges as the most critical transcriptional partner. YAP/TAZ-TEAD complexes are mainly recruited to enhancer regions and, to a certain degree, on gene promoters [90] (Figure 3). This interaction with TEAD is regarded as a promoter of oncogenic transformation [91,92]. More recently, the TEAD complex has been shown to act with the AP-1 transcriptional complex to drive target gene expression [93,94]. In addition, the YAP/TAZ-TEAD complex facilitates the recruitment of transcriptional factors BRD4, TET1, and CDK9 to enhance the liquid–liquid phase transition, allowing the transcription of a large proportion of its targets [95,96,97]. The association of YAP/TAZ with TEAD leads to chromatin remodeling and the closeness of the methyltransferase complex, allowing cancer stem cells (CSC) to acquire abnormal properties such as chemoresistance and metastasis [98]. YAP and TAZ are also binding proteins for Smads, which participate in TGFβ signaling; the binding of YAP1 and Smad7 inhibits TGFβ receptors [99]. YAP/TEAD complexes can also form to produce chemokine ligands in prostate cancer, which again aids in tumor immune evasion [100].

## 5. Role of Hippo Components and YAP/TAZ in Normal Homeostasis versus Ectopic Growth and Their Apoptotic Roles

Apoptosis is a mechanism by which cells maintain homeostasis and prevent cancer development. MST1/2 and LATS1/2 have tumor-suppressive functions and can promote apoptosis via activating pro-apoptotic signaling pathways and inhibiting anti-apoptotic pathways [98,99]. MST1/2 has been shown to activate the JNK pathway, while LATS1/2 can inhibit the anti-apoptotic protein YAP/TAZ [100]. When in the nucleus, MST1/2 leads to cell death and has downstream signaling targets such as p39 kinases, whose functions include cell differentiation and growth [80,92]. MST1/2 is also linked to the LATS kinase Ndr1/2 [101]. LATS2 is found in the centrosome; however, in response to oncogenic stress, it migrates to the nucleus and inhibits Mdm2, stabilizing p53 [102,103]. These kinases play a critical role in tissue homeostasis, and their dysregulation is correlated with cancer development. A summary of the upstream regulators of YAP/TAZ is presented in Figure 4.

YAP/TAZ are essential for organ and tissue growth during development as they act as sensors to control organ size. Their role differs from organ to organ; however, it is common in all tissues that dysregulation of YAP/TAZ leads to ectopic tissue growth and cancer.

## 6. Transgenic Expression and Conditional Knock-Out Models of YAP/TAZ

It has been demonstrated that a loss of specific components of the Hippo pathway leads to tissue overgrowth in Drosophila [9,109,110,111]. In mammals, a similar finding was noted, as YAP transgenic mice were found to have increased cell proliferation and tissue overgrowth [112,113]. Despite being considered paralogues, YAP and TAZ do not necessarily compensate each other. For instance, YAP1 knockout mice are embryonically lethal and display a shortened body axis and yolk sac abormal vasculogenesis [114]. On the other hand, TAZ knockout mice are viable with renal cysts and kidney disorders [115]. These observations have stimulated the development of tissue-specific manipulations of YAP and TAZ to delineate organ/tissue specific functional overlaps.

Liver

In the liver, an increased expression of YAP has been seen to enhance the cell proliferation of mature hepatocytes by regulating liver progenitor cells [116]. Inactivation of NF2/merlin, an upstream regulator of YAP in mammals, was found to cause the expansion of oval liver cells that are progenitors for hepatocytes and bile-duct epithelial cells. This again demonstrates the role of YAP in cell proliferation [116]. Moreover, in studies where NF2 was inactivated, deleting one YAP allele saved the liver damage previously seen in NF2-null livers [117]. Finally, in YAP knockout mice, a decline in the proliferation of liver duct cells was shown [118]. In liver cancer, it has been shown that YAP plays a role in carcinogenesis by inhibiting tumor suppressant kinases [119]. Recent findings have illustrated the complicated biology of YAP/TAZ in the liver tissues that impacts HCC development. While activation of YAP/TAZ is predominantly associated with HCC development, peri-tumoral activation of YAP/TAZ in hepatocytes suppressed tumor growth, thereby illustrating a classic case of cell competition mediated by YAP/TAZ [120]. Aside from hepatocytes, activation of TAZ in the hepatic stellate cell (HSC) population associated with myofibroblasts drives collagen-1-DDR signaling and increased stiffness to create a tumor-prone microenvironment [121].

Epidermis

In epithelial tissues, the YAP/TAZ complex regulates cell polarity and cell–cell junctions [122]. Therefore, an overabundance of YAP/TAZ has been linked with dysregulation of tissue architecture, epithelial-mesenchymal transition, and anti-apoptotic characteristics. The presence of YAP in the epidermal basal layer is linked to keratinocytes; its overexpression has been shown to cause the thickening of the layer, while its complete absence through deletion is believed to be responsible for a reduction in keratinocyte proliferation [123,124]. In the skin epidermis, contact with the ECM dependent on integrin-Src signaling is essential to activate YAP/TAZ [125].

Nervous system

As seen in the epidermis, overexpression of YAP has many negative consequences for the nervous system. One such outcome is a decrease in neural progenitor cell differentiation in cells overexpressing YAP, which is tied to the inactivation of NF2 in the dorsal telencephalon [126,127]. Interestingly, these findings are seen in animal models (mouse and chicken) through the protocadherin Fat4. In these models, the downregulation of YAP through YAP-targeting microRNAs was found to reverse the changes seen by Fat4 deficiency [128].

Pancreas

In the pancreas, overexpression of YAP leads to an expansion of ductal cells, a change in which the proliferation is maintained throughout the individual’s life [129]. In both adult human and mouse pancreas, there is a broad expression of phosphorylated MST1/2, and in tumor samples from pancreatic ductal adenocarcinoma (PDAC) patients, there is a noticeable overexpression of YAP and TAZ [130]. There is such a high overexpression that YAP has been suggested as a prognostic marker, where high amounts of nuclear YAP are linked with poor patient survival and metastasis to the liver. Furthermore, PDAC patients with high mRNA YAP expression have a 0% 5-year survival rate compared to 32% for patients with much lower mRNA YAP [131]. The poor patient outcomes are linked to the proliferative role of YAP/TAZ in PDAC cells. Moreover, YAP promotes cancer cell motility by inhibiting AKT signaling and TGFβ-YAP nuclear colocalization by PDAC cells, leading to EMT [132]. As seen in other organs, deleting even one YAP allele is sufficient to prevent this from occurring [129].

Kidney

In contrast to the organs mentioned above, in which YAP overexpression is responsible for phenotypic changes, the inactivation of YAP or TAZ in kidney precursor cells seems to lead to different outcomes. Multiple studies have demonstrated that YAP is responsible for nephron morphogenesis, whereas TAZ has a role in polycystic kidney disease [133,134].

Heart

The YAP/TAZ pathway plays a crucial role in controlling cell proliferation and organ size [135]. The study found that YAP and TAZ are essential for postnatal cardiac growth and function, and their combined deletion leads to lethal cardiomyopathy. They determined that deletion of YAP impairs neonatal heart regeneration, while activation of YAP in the adult heart improves cardiomyocyte numbers, reduces scar size, and enhances cardiac function after a heart attack [135]. Therefore, manipulating the Hippo–Yap signaling pathway could potentially promote cardiac repair after injury.

Additionally, the Wls gene is a direct target of the YAP/TAZ pathway in cardiomyocytes and plays a role in neonatal heart regeneration through Wnt signaling [136]. Several Wnt ligand genes were found to be expressed in cardiomyocytes, and reducing Wls activity disrupted noncanonical Wnt signaling in both cardiomyocytes and non-cardiomyocytes. Deletion of Wls in cardiomyocytes impaired neonatal heart regeneration after myocardial infarction (MI). Additionally, the results demonstrated that Wnt inhibitors were upregulated in cardiac fibroblasts after MI, suggesting a decrease in Wnt signaling during fibroblast activation [136].

Conditional YAP1 KO in the heart results in cardiac hypoplasia, as evidenced by reduced ventricular chamber size, ventricular septal defects, peripheral edema, and pericardial effusion [137]. While overexpressing YAP1, thereby minimizing MI injury and ameliorating cardiac function [138].

## 7. YAP/TAZ and Early Embryonic Development

Both YAP and TAZ play fundamental roles in embryonic development. Mutations that lead to a double null of both proteins prevent embryos from surviving implantation [139]. It is believed that YAP/TAZ has a critical role in the homeostasis of the body axis, neural morphogenesis, and yolk sac vasculature; therefore, in mutants of the proteins, abnormal and highly defective phenotypic changes are seen [140]. For example, YAP/TAZ is involved in cardiomyocytes’ proliferative and regenerative properties [137]. It has been reported that with YAP overexpression, an increase in the overall size of the heart is noted. Moreover, a complete deletion of YAP was shown to lead to thinning of the ventricular walls to the extent that it led to lethal heart failure. By itself, TAZ inactivation has no physiological symptoms; however, when combined with YAP inactivation, there is an increase in the symptoms listed above [33]. Consistent with the observation that inactivation of the Hippo pathway reverses cardiac failure, the proteoglycan Agrin has been shown to promote cardiac regeneration by promoting cardiomyocyte proliferation through Dystroglycan-Erk-YAP signaling [141,142]. The small percentage of defective embryos that survive to term often die shortly later from organ dysfunctions listed above (i.e., polycystic kidney disease) [133,134].

## 8. YAP/TAZ and Stem Cells

The proliferation of stem and progenitor cells is required for tissue homeostasis; therefore, they must be tightly regulated. Countless studies have discovered that YAP and TAZ are needed for the tissue regeneration role of stem cells [117,129,143]. In mouse tissue renewal, YAP and TAZ are required to maintain both the proliferation and inhibition of differentiation of transit-amplifying (TA) cells, a population of cells in which there is an abundance in the expression of both molecules [68]. Deleting YAP and TAZ can prevent both skeletal and epithelial stem cell proliferation. At the same time, the expression of YAP is required for tissue homeostasis in the liver, intestine, and mammary glands [68]. The pathway identified in this process involves the activation of the ITGA3-FAK-CDC42 signaling axis, in which Integrin α3 is inducted in TA cells, leading to the phosphorylation of YAP-S397, which in turn activates the expression of *Rheb* and potentiates mTOR signaling [68]. The role of YAP/TAZ in the maintenance of stem cells is similar in human embryonic stem cells (hESCs). Signals from TGFβ converge the SMAD pathway with the Hippo pathway to maintain pluripotency in hESCs by suppressing the expression of differentiation markers [144]. In Lgr5+ intestinal stem cells (ISCs), YAP suppresses Wnt signaling, reprograming the cells to induce regenerative properties through EGFR signaling in line with cancer initiation [145]. Altogether, these findings demonstrate the regulatory role YAP/TAZ has in the proliferation of stem cells and tissue homeostasis.

## 9. YAP/TAZ as Promoters of ‘Hallmarks of Cancer’

### 9.1. Control of Cell Proliferation and Survival

The Hippo pathway is involved in phosphorylation cascades that control tumor growth and survival. The activated Hippo pathway leads to the phosphorylation of YAP and TAZ, allowing for 14-3-3 protein-dependent YAP/TAZ cytoplasmic retention to occur and exhibiting an inhibitory effect of YAP/TAZ [106,143]. The inactivation of the Hippo kinases does not allow the phosphorylation of YAP and TAZ, thereby allowing nuclear translocation [118,146,147]. The dysregulation of YAP and TAZ leads to tumor occurrence and contributes to cancer progression [5,69]. In late-stage ovarian, liver, esophageal, and invasive breast cancers, YAP is present at a very high frequency, and TAZ is found in the nucleus of most malignant breast, lung, and colon cancers [5].

Cell growth has been linked to the interaction with TEAD co-factors; however, the role of YAP/TAZ in this process is relatively unknown. Signaling molecules such as AREG or A.X.L. and cell cycle regulators such as FOXM1 and CyclinD1 have been suggested as responsible for proliferation targets for YAP/TAZ [16,148,149]. However, other mechanisms have been suggested for YAP/TAZ-induced proliferation. The ability to bypass tissue-level checkpoints or overcome contact inhibition of YAP/TAZ may also be responsible [49]. Moreover, the downregulation of cell cycle inhibitors (p21, p27) and the expression of genes involved in PI3K/Akt and MAPK/ERK signaling pathways are theorized to be involved [150].

The YAP/TAZ complex also helps cancer cells survive under stress by inhibiting apoptosis and promoting autophagy. Anoikis, a programmed cell death occurring upon cell detachment, is believed to be repressed by YAP and TAZ through the loss of cell-substrate contacts [151]. The activation of the complex has also been shown to upregulate Bcl-2, an anti-apoptotic gene, and downregulate Bax and p53, pro-apoptotic genes [152]. Another mechanism by which tumor cells can survive and recur is K-Ras activation by YAP1, even after K-Ras inhibition [153,154]. The inactivation of cell death is not the only way cancer cells can use YAP/TAZ to evade immune detection. The activation of the complex leads to the upregulation of snail, slug, and M.M.P.s, which promote cell migration, invasion, and metastasis [155].

Moreover, cancer cells use YAP/TAZ signaling to evade current therapeutics. In the liver of mice, the overexpression of YAP has been demonstrated to block TNF-α induced cell death. At the same time, in breast cancer, the knockdown of TAZ in MCF10A mammary cells improves susceptibility to taxol and other chemotherapeutic drugs [156]. This significant finding demonstrates how YAP1 can aid tumor cell escape even with targeted chemotherapy.

### 9.2. Endowing Cancer Stem Cell Traits

YAP/TAZ have also been tied to the proliferative potential of cancer stem cells. High levels of TAZ are required for C.S.C.s to maintain their properties, such as resistance to chemotherapy, loss of differentiation markers, and self-renewal. In mouse breast cancer cells, TAZ is involved in the capacity of cells to form self-regenerating mammospheres and thereby form tumors [46,157]. In lung and breast cancers, both YAP/TAZ have been linked to cancer metastasis [23,46,158]. For this reason, YAP/TAZ is fundamental in both the formation of the tumor, the ability of the tumor to metastasize, and the maintenance of the stem cell pools that control organ size and tissue replenishment. A recent study identified an extracellular vesicle-based network (EV-Net) that was comprised of proteoglycan Agrin-enriched stem cells that effectively activated YAP to promote pancreatic ductal adenocarcinomas via lipo-protein-related receptor-4 (Lrp4) [159]. Such a specialized EV-Net network may activate YAP-induced stemness within tumor populations locally.

### 9.3. Factors Contributing to YAP/TAZ Induction in Cancer

The upregulation of YAP/TAZ leads to cellular transformations linked with human cancers; however, the factors inducing the activation of YAP/TAZ are not well understood. Many would suggest mutations in the Hippo pathway may be responsible, but aside from inherited disorders associated with NF2, mutations of Hippo pathway components are rare [2]. For this reason, it is believed that disruptions in cell polarity, loss of cell architecture, and alterations in stem cell niches may be mechanisms that blunt the Hippo cascade and induce the properties of cancer [160,161]. It is also theorized that a loss of upstream regulators such as MST1/2, LATS1/2, and NF2 and signaling pathways such as EGFR, PI4K/Akt, and RAS/RAF/MEK/ERK promote YAP/TAZ activity in cancer [162]. Environmental changes can also induce the expression of YAP/TAZ in cancer. These alterations include changes in cellular metabolism, increased glucose uptake, and hypoxia through HIF-1α, which also activates the YAP/TAZ complex in cancer [163]. In addition, a complicated TME that enriches YAP/TAZ via combined activation of oncogenic drivers and mechanical parameters may underline the existence of a feedback mechanism for a regulated function of YAP/TAZ in cancer [59,164].

### 9.4. YAP-Mediated Tumor Growth and Metastasis Are Dependent on the TEAD-Interaction Domain of YAP

The transcriptional enhancer associate domain (TEAD), which allows cells to proliferate, has been shown to bind to YAP/TAZ preferentially. Many other transcriptional factors have been shown to interact with YAP/TAZ (AP-1, TRPS1, ZEB1, etc.); however, the role of TEAD seems to be the most critical. This interaction with TEAD is regarded as a promoter of oncogenic transformation [91,92,165]. YAP/TAZ confers tumor growth and metastasis by interacting with TEAD via the TEAD-interacting domain [23,166]. The association of YAP/TAZ with TEAD leads to chromatin remodeling and the closeness of the methyltransferase complex, allowing cancer stem cells to acquire abnormal properties such as chemoresistance and metastasis [98]. YAP/TEAD complexes can also form to produce chemokine ligands in prostate cancer, which again aids in tumor immune evasion [100]. Studies have found that YAP-mediated tumor growth can be inhibited by blocking the YAP-TEAD interaction in both in vitro and in vivo models [167,168]. Interestingly, mutations in the TEAD-binding domain (T.B.D.) impair the binding of YAP to TEAD, thereby inhibiting the proliferation and metastasis of cancer [169].

### 9.5. Metastatic Potential Is Strongly Correlated with YAP/TEAD Activity

The expression of YAP has been shown to transform cells and promote epithelial-to-mesenchymal transition, enhancing tumor cells’ invasiveness and promoting metastasis [37,128,146,170]. The increased EMT potential is linked to the YAP/TEAD signaling transcription factors snail, slug, and Zeb1 [171]. In transgenic mouse models, the expression of both wild-type and mutant YAP resulted in tissue overgrowth and tumor formation [118,166,172,173]. Aside from YAP, TAZ is linked to the invasiveness of breast cancer cells, confers cancer stem cell-like properties, and interferes with proper Hippo signaling [7,23,61]. A similar finding is seen in pancreatic cancer: elevated YAP/TAZ activity increases the invasiveness of tumors. Conversely, blocking the YAP/TAZ signal blocks the metastatic potential of the cancer cells [174].

### 9.6. YAP/TAZ in the Tumor Microenvironment (TME)

The tumor microenvironment is a multifaceted environment that surrounds solid tumors and affects their growth and survival. Tumors can “hijack” many cellular systems and, as such, can change their environment into one that better suits their needs through various mechanisms in a positive feedback loop [82]. These changes include decreasing the pH to make it more acidic, creating a hypoxic environment, increasing the interstitial pressure, and increasing fibrosis [82]. Cancer-associated fibroblasts (C.A.F.s) are abundant in the TME, activated by YAP/TAZ signaling, and can promote tumor growth through the secretion of growth factors and cytokines [175]. TAZ has been shown to induce the production of inflammatory cytokines in the TME, while YAP carries out oncogenic activity because of this inflammation [119]. The complex can also polarize macrophages towards the M2 phenotype, which is much more tumorigenic and invasive [176]. YAP/TAZ can create these changes in the tumor microenvironment that account for poor patient outcomes [91,165]. In addition, the YAP mechanosensing system is also shown to be active during macrophage-mediated inflammation [177].

### 9.7. Metabolic Reprogramming and YAP/TAZ

The YAP/TAZ pathway maintains the nutrients needed for proliferation during carcinogenesis through the modulation of glycolysis, glutamine metabolism, and cholesterol synthesis [178]. This is done by reprogramming their metabolic activity so that the body produces ATP and other macromolecules needed for tumor growth [179]. It has been previously shown that cells increase their glucose demands upon YAP/TAZ activation, promoting glycolysis [180,181]. Phosphofructokinase 1 (PFK1) is believed to be fundamental for this reprogramming as it binds the transcriptional co-factors to the TEAD complex [180]. Glycolysis is not, however, the endpoint for metabolic reprogramming. YAP can reprogram glycolytic metabolism by using fatty acid oxidation, which helps increase the metastatic effects of the tumor [182].

Moreover, multiple cancers, such as leukemia, breast, esophageal, and prostate cancer, are altered by preferentially exploiting mevalonate metabolic activity. This pathway synthesizes sterols essential to tumor growth [183,184]. YAP/TAZ is believed to be controlled by this pathway through its dephosphorylation, which increases the complex accumulation in the nucleus [185].

### 9.8. Role of YAP/TAZ in Normal versus Tumor-Vasculature

During embryonic development, YAP/TAZ signaling in endothelial cells (EC) regulates angiogenesis by forming and maturing new blood vessels. The complex can be activated through multiple factors, such as mechanical forces, ECM cues, and growth factors. One example of mechanical forces is the shear stress on the vessels caused by increased blood flow, which activates transcriptional genes involved in the proliferation and migration of endothelial cells [35]. Moreover, YAP/TAZ can regulate angiogenesis by increasing pericyte recruitment by inducing PDGF-BB expression and enhancing smooth muscle cell contractility [186].

Blood vessels are fundamental for developing tumors, as they deliver oxygen and nutrients to the cells to allow them to grow. For this reason, tumor cells use angiogenesis and other neovascularization mechanisms such as vascular mimicry to form new vessels from pre-existing ones through the secretion of pro-angiogenic signals [49,187,188]. The tumor microenvironment aids in angiogenesis as it consists of cells and growth factors that create a hypoxic, acidic, and inflammatory environment; however, YAP/TAZ activation is also crucial for angiogenesis [109,146,189,190]. The expression of VEGF, a pro-angiogenic factor that promotes the growth of new blood vessels in the tumor mass, is activated through YAP/TAZ signaling [186]. VEGF signaling and VE-cadherin are also responsible for the nuclear translocation of endothelial YAP/TAZ and the remodeling of cell–cell junctions that activate the complex [2,191].

Interestingly, the outcome of YAP/TAZ signaling regarding blood vessel formation differs in normal versus tumor tissues. During development, EC-specific YAP/TAZ deletion severely impairs neoangiogenesis via impaired trafficking of VEGFR2, reflecting its role as an orchestrator of VEGF-VEGFR signaling [192]. However, in tumor tissues, YAP/TAZ signaling promotes leaky blood vessel formation, leading to hemorrhaging and impaired delivery of chemotherapeutic drugs [190]. In glioblastomas, for example, high levels of TAZ in the tumor endothelium are shown to increase blood vessel density, which is tied to the intricateness and malignancies of the tumor [190]. Furthermore, the recruitment of specific immune cells, such as tumor-associated macrophages (T.A.M.s), into the TME by YAP/TAZ signaling can also promote angiogenesis through the secretion of pro-angiogenic factors [176].

There are multiple mechanisms by which YAP/TAZ activates angiogenesis in tumor cells. One is the production of pro-angiogenic factors such as VEGF and FGF by tumor cells [186]. Another method promotes the proliferation and migration of endothelial cells to form new blood vessels [186]. As mentioned before, the YAP/TAZ complex can recruit pericytes and enhance the contractility of smooth muscle cells to increase blood flow to the tumor [186]. Furthermore, the mechanical cues (stiffness and tension) in the TME have been shown to enhance angiogenesis [186]. Agrin activation in the TME and tissue stiffness can impact VEGFR2 stability and likely maintain YAP activity [164,193].

Additionally, activation of EC-specific signaling may also trigger YAP activity. For instance, activation of STAT3 signaling was also recently shown to be essential for the nuclear localization of YAP/TAZ in ECs [194]. All these mechanisms allow the tumor to grow and migrate; therefore, targeting YAP/TAZ signaling as therapy may prevent the proliferation of cancer.

## 10. Current State of YAP1/TAZ Inhibitors and Activators

In recent years, several small-molecule inhibitors and activators of YAP/TAZ have been developed and assessed in preclinical and clinical studies.

## 11. Reagents That Inhibit the Interaction between YAP1 and TEAD

Recent experiments using Ki-Ras-driven mouse models show that YAP1 is secondarily activated down-stream of Ki-Ras. Verteporfin (VP) was initially known as a photosensitizer and is currently used in the study of the Hippo pathway. Liu-Chittenden et al. measured luciferase activity to assess the YAP1 and TEAD interaction. VP has been shown to inhibit the interaction between YAP1 and TEAD [101,195]. CA3 has been shown to suppress the growth of cancer cells in mesothelioma and esophageal adenocarcinomas. CA3 binds to the WW domain of YAP and subsequently reduces YAP expression and YAP1-TEAD complex activity [101,196]. Metformin, an AMP-activated protein kinase, has recently been proven to inhibit the YAP1-TEAD complex in solid tumors, including gliomas. A clinical trial study of metformin in rectal cancer is taking place [101].

Statins, which are safe and usually well tolerated in treating hypercholesterolemia, have been shown to inhibit YAP/TAZ signaling by blocking the mevalonate pathway. Recent studies imply that the use of these agents is associated with beneficial effects in some malignancies [197]. In addition, FAK (focal adhesion kinase) inhibitors can inactivate YAP/TAZ signaling by disrupting the interaction between YAP/TAZ and focal adhesions [198]. Similarly, ILK inhibitors have also been postulated to inactivate YAP/TAZ [69].

Moreover, VGLL4 is a transcriptional co-factor that binds to TEAD via Tondu domains. Recent studies indicate the tumor-suppressive role of VGL, and this role relies on its competition with YAP1 for binding to TEAD. Importantly, VGLL4′s tandem Tondu domains are sufficient to inhibit YAP activity [27]. In addition, studies have shown that TED-347 and its derivatives inhibit the transcriptional activity of TEAD by disrupting the protein interaction between TEAD-YAP1 [101]. Pobbati et al. identified a druggable central pocket in the YAP-binding domain of TEAD that could be targeted by NSAIDs such as Flufenamic acid, essentially blocking YAP/TEAD-dependent transcription [29].

Tang et al., in a drug screening, demonstrated VT101 and VT102 as selective TEAD palmitoylation inhibitors in NF2-deficient mesothelioma [199]. One recent study in mesothelioma showed that K-975, a new TEAD inhibitor, is a potent and highly selective inhibitor of the YAP1/TAZ-TEAD interaction [101]. On the other hand, some agents can activate YAP/TAZ. LPA (lysophosphatidic acid), a bioactive lipid, can activate YAP/TAZ signaling through the G protein-coupled receptor LPAR1 [200]. Prostaglandin E2, a lipid mediator, is another important molecule that can activate YAP/TAZ signaling through the EP4 receptor [201]. Furthermore, pharmacological inhibition of the Hippo pathway components MST1/2 and LATS1/2 can activate YAP/TAZ signaling. While the effects of these compounds have been demonstrated in preclinical studies, their efficacy and safety in human patients are still being evaluated in ongoing clinical trials. Moreover, due to the complexity of YAP/TAZ regulation and signaling, combinatorial strategies that target multiple pathways will likely be required for effective YAP/TAZ inhibition or activation.

## 12. Reagents That Modulate YAP1/TAZ-Mediated Gene Transcriptions

Several available reagents can modulate YAP/TAZ-mediated gene transcription in vitro and in vivo. Verteporfin has been shown to inhibit YAP/TAZ signaling by disrupting the interaction between YAP/TAZ and TEAD transcription factors. Verteporfin has been used in several preclinical and clinical studies to inhibit YAP/TAZ signaling in cancer cells [101,195]. Additionally, CA3 can bind to the WW domain of YAP and inhibit YAP/TEAD-dependent transcriptional activity. CA3 has been shown to inhibit the growth of several cancer cell lines in vitro and in vivo [101,196]. Moreover, some FAK inhibitors have been shown to inhibit YAP/TAZ signaling and reduce tumor growth in preclinical studies [198]. Recently, several peptide inhibitors have been developed that can block the interaction between YAP/TAZ and their binding partners, such as TEAD transcription factors (e.g., TEAD binding domain peptide), AMOT (e.g., AMOT-PDZ peptide), or angiomotin-like 2 (e.g., AMOTL2 peptide) [202].

Furthermore, some other small-molecule inhibitors of the upstream Hippo pathway components MST1/2 and LATS1/2 have been developed to activate YAP/TAZ signaling. These include XMU-MP-1 and MHY1485 [203]. These reagents can be practical tools for studying the biological functions of YAP/TAZ and developing therapeutic strategies to target YAP/TAZ-mediated signaling in cancer and other diseases. Basu et al. and Nishio et al. reported in their studies that C19 and ivermectin (and its derivative, milbemycin D) are TAZ and YAP1/TAZ inhibitors, respectively [195,204]. Kawano et al., in their study, used A549 cells, TEAD-responsive H2B-mCherry, and TAZ. They indicated that ethacridine is a TAZ activator [205].

Auto-palmitoylation of TEADs is essential for sustaining their functions [206,207]. Studies have shown that MGH-CP1 can inhibit YAP1/TAZ transcriptional function by inhibiting the auto-palmitoylation of TEAD [101]. Small-molecule inhibitors that target this lipid pocket (LP) of TEAD have been shown to impair TEAD transcription [208]. In addition, the discovery of a new class of inhibitors that bind to the palmitate binding pocket (PBP) of TEAD to induce side-binding to hydrophobic residues to impair TEAD-dependent transcription further raises exciting possibilities for targeting the YAP/TEAD pathway [209]. Therefore, inhibitor screens that target the PBP pocket of TEAD may also serve as an essential platform for switching the YAP-TEAD transcriptional program vital for tissue regeneration [210].

Additionally, pazopanib and dasatinib as tyrosine kinase inhibitors have been demonstrated to result in the phosphorylation and degradation of YAP1/TAZ and, subsequently, their cytoplasmic sequestration [101]. Mechanical modulation of YAP/TAZ can be observed when cancer cells are grown in a soft versus rigid ECM. This suggests that loss of tissue architecture and altered ECM composition can activate YAP/TAZ [33]. In this context, agrin-dependent mechanical changes to YAP were also shown to be blocked by a function-blocking antibody [63].

Although TEADs play an essential role in YAP/TAZ function, there are other transcription factors known to interact with the WW domains of YAP/TAZ, including RUNX, PPARγ, Pax3, TBX5, and TTF-1 for TAZ [211,212,213,214,215] and Smad1, Smad2/3, RUNX, ErbB4, and p73 for YAP [38,216]. In contrast with TEAD, these transcription factors contain a PPXY motif, which is considered essential for recognition by YAP/TAZ [36]. Smad proteins are transcriptional regulators through which TGF-β regulates many developmental events in human embryonic stem cells (hESCs) [36]. Studies show that YAP controls Smad nuclear localization and coupling to the transcriptional machinery [17]. Angiomotin family proteins (AMOT) were identified as “angiostatin binding proteins”, accelerating endothelial cell migration and angiogenesis. Lately, YAP/TAZ and AMOT have been demonstrated to interact, resulting in YAP/TAZ inhibition through various mechanisms [36]. Recent studies have shown that the topoisomerase inhibitor A35 regulates the transcription of YAP target genes associated with growth and cycle regulation, resulting in G2/M arrest and growth inhibition. The mechanism of action is believed to be decreasing YAP1 nuclear localization and decreasing YAP/TAZ expression and its target genes [217,218].

Jiao et al. suggested a new peptide to inhibit YAP/TEAD interaction and transcription of its target genes. “Super-TDU”, such as VGLL4, competes with YAP in binding to TEADs, resulting in the inhibition of the oncogenic activity of YAP and down-regulated expression of target genes CTGF, CYR61, and CDX2 [27,82,219]. Gibault et al. indicated that Verteporfin (VP) remarkably inhibits the luciferase activity of TEAD and decreases YAP/TAZ expression across MDA-MB-231 cell lines in breast cancer. This result is related to reducing the co-transcriptional function of the YAP/TAZ/TEAD complex [82,220]. According to studies, VP can induce the cytoplasmic retention of YAP and, indirectly, TAZ through increased amounts of 14-3-3σ protein in endometrial cancer tissues. Further, VP multiplies p53 levels, an instrumental process for the potentiation of its effect [221].

## 13. Reagents That Modulate the Subcellular Localization of YAP1/TAZ

Several reagents can modulate the subcellular localization of YAP1/TAZ, including soluble growth factors, ECM proteins, and receptor tyrosine kinases (RTKs) and their ligands. The subcellular localization of YAP1/TAZ correlates with their function. In the canonical Hippo pathway, unphosphorylated YAP1 and TAZ locate in the nucleus and regulate transcription, whereas phosphorylated YAP1 and TAZ degrade in the cytoplasm via the proteasome [195]. Bao et al. [222] used U2OS cells expressing GFP-YAP1 and showed that dobutamine drags GFP-YAP1 from the nucleus to the cytoplasm. Jang et al. [223] and Park et al. [224] identified new TAZ modulators (TM-53, TM-54, and TM-25659) by using COS cells that express GFP-TAZ.

Statins such as cerivastatin and simvastatin induce YAP1/TAZ phosphorylation and inhibit their nuclear translocation, thereby affecting the expression of their target genes. This inhibition effect could be strongly reinforced when statins are combined with dasatinib [101]. New researchers have found that ECM stiffness can inhibit the Salvador (Sav)–LATS association, which inhibits the Hippo signal pathway [71,225]. Al and Src can increase YAP activity by phosphorylating LATS or directly phosphorylating YAP [88,226]. Controlling the Hippo pathway by mechanical upstream signals, including ECM stiffness, mainly depends on the effects of mechanical signals on the F-actin cytoskeleton, tension within the cytoskeleton, cell–cell and cell–matrix adhesions, cell shape, and cell density [227]. Myosin contraction grade is usually proportional to the activity of YAP [225]. The Rho-Rock (Rho-associated protein kinase)-MLC (non-myosin II light chain) pathway is one of the primary regulators of the cytoskeleton [151,228]. By inhibiting the main cytoskeleton regulatory factor, Rho can inhibit the consequences of the cytoskeleton-mediated mechanical signals on the Hippo pathway [49,151,229].

Calvo et al. have studied YAP in the context of the mechanical activation of cancer-associated fibroblasts (CAFs) [60]. Here, YAP is activated by ECM stiffening to stimulate several CAF protumorigenic features [33]. Matrix stiffness-based induction of YAP is a remarkable feature of CAFs that sustain YAP-actomyosin contractility in a feed-forward loop [60]. Surprisingly, YAP self-sustains its activity in CAFs by increasing the expression of the myosin regulatory light chain gene (MYL9) and actin contractility, resulting in “inside-out” ECM stiffness. In this regulation, mechanotransduction happens independently of LATS-mediated phosphorylation and MST1/2, while it requires Rock, myosin, and Src activity [33].

Guan et al. showed that YAP and TAZ are nuclear and active in cells growing in low-density spaces but become inactive in confluent cultures [15]. Cell-cell contact partly cripples YAP/TAZ function through the Hippo pathway but leaves cells with enough nuclear YAP/TAZ collection to continue proliferation. A second mechanical inhibitory phenomenon must occur for noticeable growth arrest and quantitative YAP/TAZ nuclear exclusion [35]. The main components of adhesive junctions are cadherins, which bind cells to each other in different tissues [35]. A-Catenin connects cadherin in cells. 14-3-3 proteins build a complex with α-Catenin and phosphorylated YAP. In an in vitro study on MCF-7 and MCF-10A, it was shown that, as a result, YAP activity is suppressed by retaining YAP in the cytoplasm, causing the inactivation of YAP [123,230]. In a study of breast cancer in MCF10A cells, researchers showed that protein-tyrosine phosphatase type 14 (PTPN14), another component of AJ, interacts with YAP through its PPXY motif and the WW domain of YAP. This interaction results in the inactivation of YAP [231,232]. Disrupting the activity of Ajs or TJs causes the activation of YAP/TAZ in different cells [54].

In addition, the Hippo pathway is modulated by some extracellular ligands or growth factors and multiple signaling pathways. Epidermal growth factor (EGF) through the EGF-receptor (EGFR), phosphoinositide 3-kinase (PI3K), and phosphoinositide-dependent kinase (PDK1) can turn off the Hippo pathway [233]. In addition, YAP/TAZ can be activated with EGFR by Ras-Raf-Mitogen activated kinases (MAPK) signaling [234]. Amphiregulin (AREG) is a growth factor that triggers EGFR and consequently activates YAP/TAZ [16].

Extracellular matrix (ECM) is related to focal adhesions (F.As) in cells. Integrins in focal adhesions can activate YAP/TAZ through Rho-GTPases [28]. Furthermore, integrins turn off Hippo signaling to activate YAP/TAZ by inactivating Merlin [235]. However, thousand-and-one amino acid kinases (TAOK1/2/3) are demonstrated to phosphorylate and trigger Hippo pathway core component MST1/MST2 kinases [52,236].

Many cancer-associated signaling networks engage in regulatory crosstalk with the Hippo pathway, often at the level of the YAP and TAZ proteins. The WNT, transforming growth factor-β (TGFβ)–bone morphogenetic protein (BMP), Hedgehog (HH), Notch, insulin, and mTOR pathways have all been found to interact with the Hippo pathway. Recent data suggest that somatic mutations in the proteins of these other signaling networks cause YAP and TAZ hyperactivation even in the presence of a wild-type Hippo pathway tumor suppressor network. For example, increased WNT pathway signaling through activated β-catenin in colorectal cancer could activate YAP by physically interacting with YAP, increasing its nuclear accumulation, and driving YAP1 transcription, thus causing increased YAP protein expression [2]. Overall, these reagents’ modulation of YAP1/TAZ subcellular localization represents a promising approach for developing novel cancer therapies.

## 14. Reagents That Induce TAZ-Dependent Sphere Formation

Various reagents can induce TAZ-dependent sphere formation in MCF10A cells. LPA (lysophosphatidic acid) is a bioactive lipid that promotes TAZ-dependent sphere formation in MCF10A cells. LPA activates RhoA and its downstream effector ROCK, which leads to the activation of TAZ and the induction of sphere formation [199].

EGF is a growth factor that induces TAZ-dependent sphere formation in MCF10A cells. EGF activates the EGFR receptor and downstream signaling pathways, including the PI3K-Akt and MEK-ERK pathways, which induce TAZ’s nuclear localization and activation [237].

*Y-27632:* Y-27632 is a small inhibitor of ROCK that promotes TAZ-dependent sphere formation in MCF10A cells. By inhibiting ROCK, Y-27632 prevents TAZ’s phosphorylation and cytoplasmic retention, allowing it to translocate to the nucleus and induce sphere formation [238].

Overall, the induction of TAZ-dependent sphere formation in MCF10A cells by these reagents can be used for further studies to identify novel therapeutic targets. According to some studies, suppressing YAP1 and TAZ can inhibit in vitro tumor sphere formation, 3D-matrigel growth, and migration of special cancer cells [195]. In addition, in gastric cancer, VP has been shown to inhibit the expression of YAP1/TAZ-TEAD, suppressing stem cell sphere formation and tumorigenesis [101].

## 15. Implications on Immunotherapy

Activation of YAP/TAZ can promote carcinogenesis by reducing apoptosis and increasing cell proliferation. This pathway is frequently active in several cancer types, including liver, lung, and breast cancer, and it is linked to a poor prognosis, metastasis, and treatment resistance [96]. The YAP/TAZ pathway is crucial in regulating the immune response in cancer. For example, activation of the YAP/TAZ pathway has been shown to inhibit the function of immune cells, such as T cells and natural killer cells, which are essential for recognizing and attacking cancer cells [159].

Immune checkpoint inhibitors enhance the immune system’s ability to recognize and attack cancer cells. In preclinical models, inhibiting the YAP/TAZ pathway has been shown to enhance the efficacy of immunotherapies such as immune checkpoint inhibitors, which block proteins that inhibit the immune response [168]. However, some cancers can evade the immune response and suppress the activity of immune cells, leading to resistance to immunotherapy. Cancers can evade the immune response through YAP/TAZ-mediated suppression of immune cell activity [168].

CXCL2 and CXCL1 are chemokines secreted by YAP/TAZ-high tumors that recruit and activate MDSCs and TAMs [239]. These cells suppress the activity of CD8+ T cells, which are essential for anti-tumor immune responses. YAP/TAZ-high tumors also secrete CCL2 and recruit TAMs to the tumor microenvironment, which then produce IL6 and GM-CSF to further amplify the accumulation of MDSCs and inhibit the activity of CD8+ T cells from generating an immunosuppressive tumor microenvironment. Collectively, the secretion of these agents enables evasion of immune surveillance and highlights the importance of therapeutic targeting of YAP/TAZ to improve anti-tumor immune responses.

Overall, the YAP/TAZ pathway is an important target for cancer therapy. There is growing interest in exploring the potential of YAP/TAZ inhibitors in combination with immunotherapies for cancer treatment. A summary of these targets is presented in Figure 5.

## 16. YAP/TAZ Inhibition for Cancer Immunotherapy

The YAP/TAZ pathway can be used to increase the effectiveness of PD-1 therapy by reducing the suppression of immune cell activity [169]. PD-1 is a type of immune checkpoint receptor that regulates immune cell activation. In some cancers, activation of the YAP/TAZ pathway can lead to increased expression of PD-1 on the surface of immune cells, contributing to resistance to anti-PD-1 therapy [82,239]. These immune checkpoint receptors act as a “shield” for cancer cells, protecting them from the immune system’s attack. When exposed to PD-L1, the immune system signals immune cells to stop their attack, suppressing the immune response. Activating the YAP/TAZ pathway can also increase the expression of other immune checkpoint receptors, such as CTLA-4 and TIM-3, which have similar effects on the immune response to cancer [170]. By increasing the expression of these immune checkpoint receptors, cancer cells can evade the immune response and become resistant to immunotherapy. Several YAP/TAZ inhibitors, such as verteporfin, have been shown to inhibit the growth of tumors and enhance the anti-tumor immune response in preclinical models. YAP/TAZ inhibitors can increase the infiltration and activation of T cells within the tumor microenvironment, which is critical for the success of immune checkpoint blockade therapies.

Targeting the YAP/TAZ pathway has been shown to reverse the increased expression of immune checkpoint receptors and increase the efficacy of immunotherapy [170]. Several YAP/TAZ inhibitors, such as verteporfin, have been shown to inhibit the growth of tumors and enhance the anti-tumor immune response [150]. YAP/TAZ inhibitors can increase the infiltration and activation of T cells within the tumor microenvironment, which can be critical for the success of immune checkpoint blockade therapies [240]. YAP/TAZ inhibition has also been shown to reduce the expression of PD-1 on the surface of immune cells and enhance the efficacy of anti-PD-1 therapy in preclinical studies [171]. This suggests that targeting the YAP/TAZ pathway may be a promising strategy for improving the effectiveness of anti-PD-1 therapy in patients with cancer. Studies have shown that activation of the YAP/TAZ pathway in hepatocellular carcinoma (H.C.C.) can suppress the immune response to cancer, reducing the efficacy of immunotherapy [172]. The YAP/TAZ pathway has been shown to inhibit the activation and proliferation of T cells [173], which are critical for recognizing and attacking cancer cells. By suppressing T cell function, the YAP/TAZ pathway can limit the immune system’s ability to respond to cancer. Preclinical studies have demonstrated that targeting the YAP/TAZ pathway can reverse the suppression of the immune response to HCC [174]. For example, YAP/TAZ inhibition has been shown to enhance T-cell activation and proliferation and increase the sensitivity of H.C.C. cells to immunotherapy. By increasing the immune response to cancer, targeting the YAP/TAZ pathway can potentially improve the efficacy of immunotherapy for H.C.C. It is important to note that while these preclinical studies provide promising results, further research is needed to fully understand the role of the YAP/TAZ pathway in the efficacy of immunotherapy for H.C.C. and to translate these findings into clinical practice.

Studies have shown that activation of the YAP/TAZ pathway in breast cancer can suppress the immune response to cancer, making it less responsive to immunotherapy [175]. Preclinical studies on the role of the YAP/TAZ pathway in breast cancer have demonstrated its impact on the efficacy of immunotherapy. Activation of the YAP/TAZ pathway has been linked to the suppression of the immune response to cancer in breast cancer, making it less responsive to immunotherapy [241]. In this study, the authors investigated the role of YAP in breast cancer cells undergoing TGF-β1-induced apoptosis and epithelial-mesenchymal transition (EMT), both of which have been implicated in the resistance of cancer cells to immunotherapy. The authors found that YAP activation inhibited TGF-β1-induced apoptosis and EMT by upregulating epidermal growth factor receptor (EGFR) expression. They also showed that targeting YAP using small molecule inhibitors sensitized breast cancer cells to TRAIL-induced apoptosis and enhanced immunotherapy’s efficacy in a mouse breast cancer model. The authors concluded that YAP inhibition might be a promising strategy for overcoming resistance to immunotherapy in breast cancer.

The YAP/TAZ pathway can upregulate PD-L1 expression, which has implications for the effectiveness of immunotherapy, particularly anti-PD-1 therapy. Targeting the YAP/TAZ pathway can reduce the expression of PD-L1 on the surface of cancer cells and increase their sensitivity to anti-PD-1 therapy. Further research is needed to fully understand the potential of YAP/TAZ inhibition as a strategy for enhancing the efficacy of anti-PD-1 therapy in cancer patients.

## 17. Modulation of Cells of the Adaptive Immune System

The YAP/TAZ pathway can modulate adaptive immune system cells by regulating genes involved in immune cell activation, proliferation, and survival [159,169]. The YAP/TAZ pathway can also modulate the expression of cytokines and chemokines that influence immune cell trafficking and recruitment to the tumor microenvironment.

YAP and TAZ are transcriptional co-activators that bind to target genes and activate their expression [242]. For example, YAP/TAZ has been shown to regulate the expression of interleukin-2 (IL-2), an essential cytokine for T-cell activation and proliferation [243]. By modulating the expression of IL-2, the YAP/TAZ pathway can play a role in the activation and expansion of T cells, which are critical players in the adaptive immune response to cancer. In addition, the YAP/TAZ pathway has also been shown to regulate the expression of other genes involved in T cell function. One is CD28, a co-stimulatory molecule essential for T cell activation and survival [179]. By modulating the expression of CD28, the YAP/TAZ pathway can play a role in the survival and function of T cells, which can impact the effectiveness of immunotherapy.

## 18. Modulation of T-Cells

T cells, also known as T lymphocytes, are white blood cells that play a central role in the adaptive immune response to cancer. They are responsible for recognizing and attacking cancer cells through T-cell activation. Activated T cells produce cytokines and release cytotoxic molecules that kill cancer cells. However, the activation and proliferation of T cells can be suppressed by the YAP/TAZ pathway, which has been shown to inhibit T cell activation and reduce T cell proliferation in preclinical studies [169,180]. This suppression of T cell activation and proliferation can lead to a decreased immune response to cancer, thereby reducing the efficacy of cancer immunotherapy. Targeting the YAP/TAZ pathway has been shown to reverse this suppression of T cell activation and proliferation, as demonstrated in preclinical studies. For example, YAP/TAZ inhibition has been shown to enhance T cell activation and proliferation, leading to increased T cell-mediated immune responses to cancer [169]. Additionally, YAP/TAZ inhibition has been shown to increase the sensitivity of cancer cells to immunotherapy by enhancing the ability of T cells to recognize and attack cancer cells.

Growing evidence suggests that the YAP/TAZ pathway may play a role in regulating T-cell exhaustion, a state of dysfunction and diminished function that occurs in T-cells during chronic viral infections and cancer [244,245]. Studies have shown that the YAP/TAZ pathway is activated in exhausted T-cells and that this activation may contribute to the development of T-cell exhaustion by promoting the expression of inhibitory receptors such as PD-1 and TIM-3, which are associated with T-cell dysfunction [240]. Furthermore, YAP/TAZ inhibition has been shown to improve T-cell function and reduce T-cell exhaustion in mouse models of chronic viral infection and cancer [240]. More research is needed to fully understand the relationship between the YAP/TAZ pathway and T-cell exhaustion, and it may represent a promising strategy for reversing T-cell exhaustion and improving immune function in chronic viral infections and cancer.

In conclusion, the YAP/TAZ pathway can modulate T cells, suppressing the immune response to cancer. Targeting the YAP/TAZ pathway has the potential to reverse this suppression and enhance the immune response to cancer. Thus, it may be a promising strategy for improving the efficacy of cancer immunotherapy. Further research is needed to fully understand the mechanisms by which YAP/TAZ modulates different types of adaptive immune cells and their implications for immunotherapy.

## 19. YAP/TAZ Pathway about Hypoxia and Drug Resistance in Cancer Immunotherapy

In cancer, dysregulation of the YAP/TAZ pathway has been implicated in the development of drug resistance and tumor progression. The YAP/TAZ pathway can modulate the tumor microenvironment by responding to hypoxia. Hypoxia, or low oxygen levels, is a common feature of solid tumors and can activate the YAP/TAZ pathway, leading to the expression of genes that promote tumor cell survival and proliferation [39]. In addition, hypoxia can also induce the expression of immune checkpoint proteins, such as PD-L1, which can inhibit the activity of immune cells and contribute to immune evasion by the tumor [181].

Furthermore, YAP/TAZ pathway activation has been linked to drug resistance in cancer. For example, studies have shown that YAP/TAZ activation can lead to the upregulation of drug efflux pumps, decreasing the accumulation of chemotherapy drugs within the tumor cells and reducing their effectiveness [170]. In addition, YAP/TAZ activation has also been associated with the upregulation of anti-apoptotic genes, which can prevent tumor cells from undergoing cell death in response to chemotherapy. Possible therapeutic targeting of the YAP/TAZ/TEAD as a single agent or a combination approach is presented in Figure 6.

## 20. Convergence between the Gut Microbiota, Immunotherapy and the Hippo Pathway

Several papers have indicated a complex interplay between gut microbiota and immunotherapy [246,247,248,249,250]. The gut microbiome plays a crucial role in modulating the immune system, and its composition can impact the efficacy of cancer immunotherapy. Certain bacteria in the gut can promote an anti-tumor immune response, while others can inhibit it. Thus, manipulating the gut microbiome may enhance the response to immunotherapy. Gopalakrishnan et al. [251,252] describe a study in which germ-free mice (i.e., those lacking a gut microbiota) had poorer responses to cancer immunotherapy than mice with a normal microbiome. This suggests that the gut microbiome plays a role in enhancing the efficacy of cancer immunotherapy. Additionally, the authors report that certain bacterial species, such as *Bifidobacterium spp.* and *Akkermansia muciniphila*, were more abundant in patients who responded well to immunotherapy than non-responders.

The composition of the gut microbiome can also influence the development of certain cancers and may impact the response to other cancer therapies beyond immunotherapy. For example, Routy et al. [253] conducted a study in non-small cell lung cancer patients who received anti-PD-1 immunotherapy. They found that patients who responded to the treatment had more *Akkermansia muciniphila* and other bacteria in their gut microbiome. In comparison, non-responders had a lower abundance of these bacteria. The authors also showed that the gut microbiome was necessary for the efficacy of immunotherapy, as fecal microbiota transplantation (FMT) from responders to non-responders improved treatment response in the latter group.

The Hippo pathway has recently been implicated in the pathogenesis of inflammatory bowel disease (IBD) [254]. As detailed earlier, the Hippo signaling pathway is crucial in organ size control, tissue homeostasis, and cancer development, and it thus may also be involved in the pathogenesis of IBD. This is an essential connection given that the Hippo pathway is dysregulated in the inflamed intestinal mucosa of patients with IBD, with increased YAP/TAZ activity and expression of downstream target genes [255]. Moreover, genetic or pharmacological manipulation of the Hippo pathway in animal models of IBD has been shown to affect the severity of the disease, suggesting that this pathway may be a potential therapeutic target. Recent studies have indicated an association between abnormal mismatch repair (MMR) genes and the Hippo pathway in pediatric malignancies. MMR deficient genes have been shown to activate the Hippo pathway, causing increased proliferation in tumor cells [2,256,257,258]. It has been shown that activation of the Hippo signaling pathway in children can cause cancers such as hepatoblastoma, osteosarcoma, neuroblastoma, and rhabdomyosarcoma [259,260]. Recent reports suggest that disruption in the Hippo pathway, predominantly YAP upregulation, is seen in both alveolar and embryonal rhabdomyosarcoma. YAP is involved in neural crest phenotype regulation and migration. During neural crest differentiation and maturation, YAP expression decreases. Studies have shown YAP and TAZ overexpression in neuroblastoma and their correlation with negative outcomes [260]. Furthermore, in vitro studies suggest that TAZ and YAP activation can lead to hepatoblastoma tumor growth [259].

Although a clear linkage between the Hippo pathway, gut microbiota, and immunotherapy has not been established, there may be potential areas of convergence that can be leveraged to provide multi-modal effectiveness of immunotherapy in treating cancer. This hypothetical convergence is presented in Figure 7.

## 21. Epigenetic Regulation on the Hippo Pathway

Epigenetic regulation can play an important role in controlling the activity of the Hippo pathway. For example, studies have shown that DNA methylation and histone modifications can influence the expression of genes that are involved in the Hippo pathway. One study found that DNA methylation of the promoter region of the YAP1 gene, which encodes the YAP protein, was associated with decreased expression of YAP in breast cancer cells [261]. YAP is a key downstream effector of the Hippo pathway, and its expression is tightly regulated to ensure proper control of cell proliferation and survival [262,263].

In addition, histone modifications can also affect the activity of the Hippo pathway. For example, histone deacetylase inhibitors (HDACi) could promote the expression of the Hippo pathway genes MST1 and LATS2 in breast cancer cells [264]. HDACi inhibits the activity of histone deacetylases, which normally remove acetyl groups from histones and thereby reduce gene expression. By inhibiting histone deacetylases, HDACi can increase the acetylation of histones, which is associated with increased gene expression [265].

Epigenetic regulation can also be influenced by environmental factors, such as diet and stress. For example, a high-fat diet could lead to hypermethylation of the promoter region of the LATS2 gene in mice, which resulted in decreased expression of LATS2 and activation of the Hippo pathway [266,267]. Overall, these findings suggest that epigenetic regulation can play an important role in controlling the activity of the Hippo pathway. Understanding the mechanisms by which epigenetic modifications influence the activity of the Hippo pathway may lead to the development of new therapies for diseases such as cancer.

YAP/TAZ-TEAD oncoproteins work by binding to specific DNA sequences and activating the expression of genes that promote cell proliferation and survival [80,268]. Recent research has suggested that epigenetic regulation may play an important role in controlling the activity of YAP/TAZ-TEAD oncoproteins. The activity of YAP/TAZ-TEAD oncoproteins can be modulated by changes in DNA methylation, histone modifications, and non-coding RNA expression [35,269,270,271]. One study found that DNA methylation of the promoter region of the YAP1 gene, which encodes the YAP protein, was associated with decreased expression of YAP in breast cancer cells [261]. Another study found that histone acetylation of the TEAD4 gene, which encodes the TEAD protein, was associated with increased expression of TEAD in ovarian cancer cells [272,273]. Other reports have suggested that non-coding RNAs, such as microRNAs, may also play a role in regulating the activity of YAP/TAZ-TEAD oncoproteins. For example, the microRNA miR-145 was able to inhibit the activity of YAP in breast cancer cells [271,274,275].

Overall, these findings suggest that epigenetic regulation may play an important role in controlling the activity of YAP/TAZ-TEAD oncoproteins in cancer cells. The mechanisms by which epigenetic modifications influence the activity of these oncoproteins may lead to the development of new therapies for cancer treatment. The interaction between diet, gut microbiome, exercise, and epigenetic modifications through the YAP/TAZ pathway is an area of active research, and there is still much to be learned about this complex interplay. YAP/TAZ are transcriptional co-activators that play a role in cellular proliferation and differentiation, and they have been shown to be modulated by various environmental factors, including diet, exercise, and gut microbiota. Several studies have suggested that YAP/TAZ activity may be influenced by the composition of the gut microbiome, which in turn can be modulated by diet and exercise [276]. However, the specific mechanisms by which diet, gut microbiome, and exercise interact with the YAP/TAZ pathway to induce epigenetic modifications are still not fully understood. Further research is needed to elucidate the complex signaling pathways involved and to determine the specific effects of different dietary and exercise interventions on the gut microbiome and epigenetic modifications mediated by YAP/TAZ. Overall, while there is evidence to suggest that diet, gut microbiome, and exercise can influence YAP/TAZ-mediated epigenetic modifications, the exact nature of this interaction and its implications for human health remain an area of active investigation.

## 22. Conclusions and Future Directions

Given the critical role of the YAP/TAZ pathway in modulating the tumor microenvironment and promoting drug resistance in cancer, targeting this pathway has emerged as a potential therapeutic strategy for cancer immunotherapy. Further research is needed to fully understand the complex interactions between the YAP/TAZ pathway, the tumor microenvironment, and cancer immunotherapy and develop effective therapeutic strategies for cancer treatment.

## Figures and Tables

**Figure 1 cancers-15-03468-f001:**
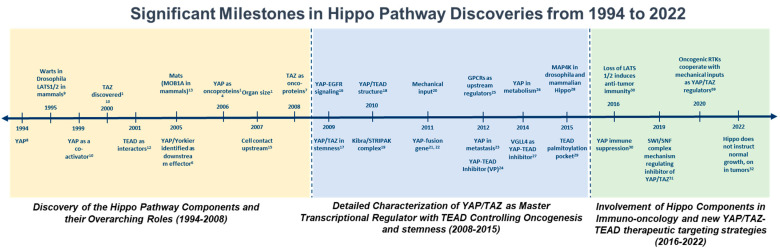
Significant Milestones in Hippo Pathway Discoveries from 1994 to 2022. The Hippo pathway has been a critical regulator of various biological processes for three decades, including organ size regulation, stem cell maintenance and differentiation, tumor suppression, tissue regeneration, and immune responses. Since the 2000s, dysregulation of the Hippo pathway has been implicated in cancer progression. Over the last decade, research efforts have focused on therapeutically targeting this pathway to treat cancer, with most recent advances focusing on the role of this pathway in enhancing immunotherapy efficacy [1,6,7,8,9,10,11,12,13,14,15,16,17,18,19,20,21,22,23,24,25,26,27,28,29,30,31,32]. Abbreviations: Epidermal growth factor receptor, EGFR; G-protein coupled receptors, GPCRs; large tumor suppressor 1/2, LATS1/2; mitogen-activated protein kinase kinase kinase kinase 4, MAP4K4; MOB Kinase Activator 1A, MOB1A; receptor tyrosine kinases, RTKs; striatin (STRN)-interacting phosphatase and kinase, STRIPAK; SWItch/Sucrose Non-Fermentable, SW1/SNF; transcriptional enhanced associate domain, TEAD; transcriptional co-activator with PDZ-binding motif, TAP; Vestigial like family member 4, VGLL4; Yes-associated protein, YAP.

**Figure 3 cancers-15-03468-f003:**
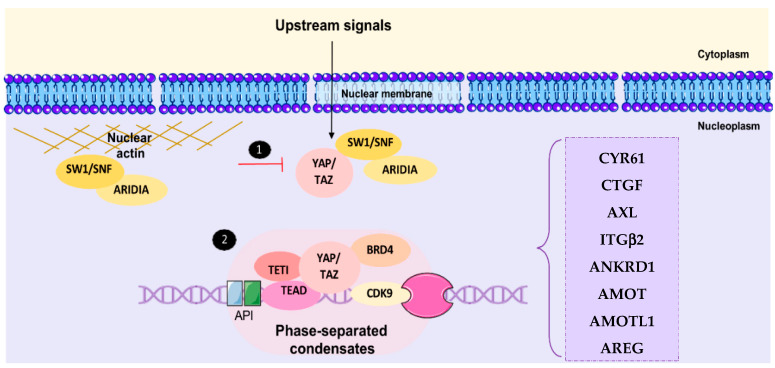
Regulation of gene expression by YAP/TAZ within the nucleus. (1) Nuclear actin, SWI/SNF chromatin remodeling complexes, and ARIDIA regulate YAP/TAZ by promoting their nuclear localization, association with chromatin, and activation of target genes. SWI/SNF and ARIDIA control DNA accessibility, while nuclear actin enhances transcriptional activity. These factors are critical for YAP/TAZ-mediated transcriptional regulation and oncogenic transformation. (2) When the YAP/TAZ complex is translocated into the nucleus, it forms a complex with DNA-binding partners such as TEADs to activate the transcription of target genes, including CTGF, CYR61, ANKRD1, and AREG. In addition to these genes, the YAP/TAZ-TEAD complex can also activate the transcription of other genes such as CYY61, AXL, ITGB2, AMOT, AMOTL1, and ADAMTS1, which are involved in a variety of cellular processes, including angiogenesis, cell migration, and ECM remodeling. More recently, the TEAD complex has been shown to act with the AP-1 transcriptional complex to drive target gene expression. The YAP/TAZ-TEAD complex facilitates the recruitment of transcriptional factors, chromatin remodeling, and the recruitment of transcriptional factors BRD4, TET1, and CDK9 to enhance liquid-liquid phase transition, allowing the transcription of a large proportion of its targets. This interaction with TEAD is considered a promoter of oncogenic transformation and can lead to abnormal properties in cancer stem cells, such as chemoresistance and metastasis [35,80,90,91,92,93,94,95,96,99]. Abbreviations: Actin-related proteins in the nucleus, ARIDIA; Activator protein-1; A Disintegrin and Metalloproteinase with Thrombospondin Motifs 1, ADAMTS1; Amphiregulin, AREG; Angiomotin, AMOT; Angiomotin-like protein 1, AMOTL1; Ankyrin repeat domain-containing protein 1, ANKRD1; AXL receptor tyrosine kinase, AXL; Bromodomain-containing protein 4, BRD4; Chondroadherin-like protein, CYY61; Connective tissue growth factor, CTGF; Cyclin-dependent kinase 9, CDK9; Cysteine-rich angiogenic inducer 61, CYR61; Extracellular matrix, ECM; Integrin beta-2, ITGb2; Switch/Sucrose non-fermentable, SWI/SNF; Ten-eleven translocation 1, TET1; Transcriptional co-activator with PDZ-binding motif, TAZ; Transcriptional enhanced associate domains, TEADs; Yes-associated protein, YAP.

**Figure 4 cancers-15-03468-f004:**
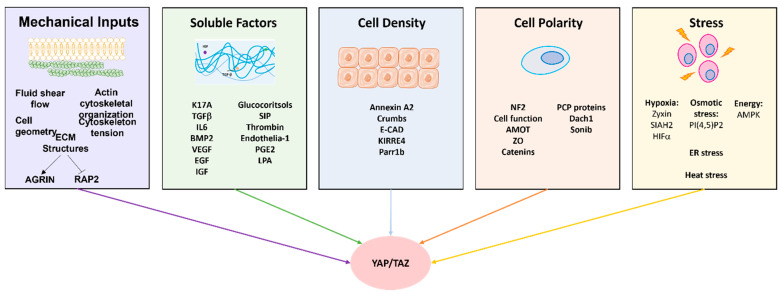
Upstream regulators of YAP/TAZ. A complex interplay between various signaling pathways, cell density, cell polarity, stress, and mechanical inputs regulates YAP/TAZ activity. Soluble factors such as growth factors, cytokines, and hormones activate YAP/TAZ through various signaling pathways. YAP/TAZ activity is also regulated by cell density and cell polarity. YAP/TAZ promotes proliferation at low cell densities and differentiation at high cell densities. YAP/TAZ is inhibited in polarized cells, whereas in non-polarized cells, it is active. Additionally, various stresses such as oxidative stress, DNA damage, and mechanical inputs such as substrate stiffness and cell stretching can activate YAP/TAZ through different pathways [101,102,103,104,105,106,107,108]. At low ECM stiffness, active RAP2 promotes Rho GTPase activating protein 29 (ARHGAP29) and mitogen-activated protein kinase (MAP4K4/6/7), which in turn results in YAP/TAZ inactivation [73] Abbreviations: AMP-activated protein kinase, AMPK; Annexin A2, ANXA2; Apoptosis-associated speck-like protein containing a CARD, ASIN; Bone morphogenetic protein 2, BMP2; Crumbs homolog 1, CRB1; Dachshund homolog 1, Dach1; Epidermal growth factor, EGF; Hypoxia-inducible factor alpha, HIFα; Insulin-like growth factor, IGF; Interleukin-6, IL6; Kinase suppressor of Ras 1, KSR1; Kin of IRRE-like protein 4, KIRREL4; Lysophosphatidic acid, LPA; Neurofibromin 2, NF2; Phosphatidylinositol 4,5-bisphosphate, PI(4,5)P2; Protein associated with Rho, RAP2; Protein kinase A, PKA; Protein kinase C, PKC; Protein kinase D, PKD; Protein kinase LKB1, LKB1; Prostaglandin E2, PGE2; Par-3 family cell polarity regulator 1B, PARD1B; PCP (planar cell polarity) proteins, PCP; Siah E3 ubiquitin protein ligase 2, SIAH2; Stress-induced phosphoprotein 1, STIP1; Transforming growth factor-beta, TGF-β; Vascular endothelial growth factor, VEGF; Zonula occludens, ZO.

**Figure 5 cancers-15-03468-f005:**
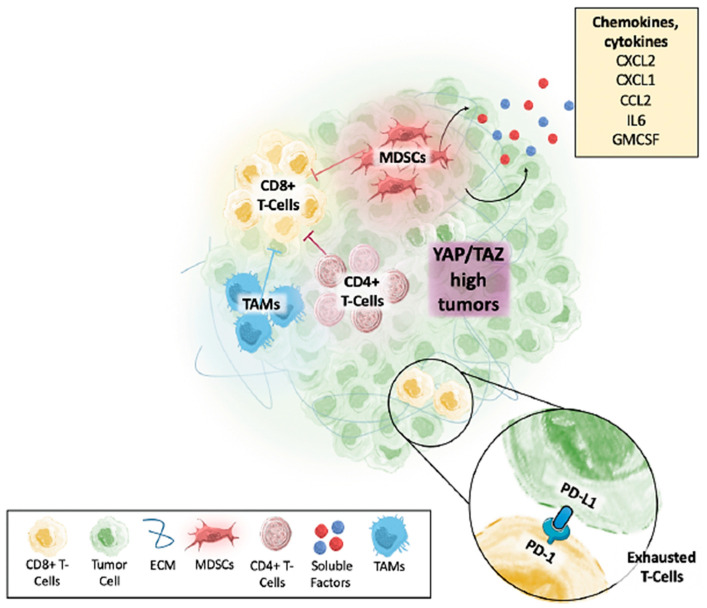
Overview of immune surveillance by YAP/TAZ. YAP/TAZ is known to play a role in the immune surveillance of tumors. In YAP/TAZ high tumors, the transcription co-activators YAP and TAZ are associated with decreased infiltration of CD8+ T cells and increased accumulation of immunosuppressive cells such as MDSCs and TAMs. This is believed to be due to the secretion of chemokines and cytokines such as CXCL2, CXCL1, CCL2, IL6, and GM-CSF, which recruit and activate these immunosuppressive cells. These cells then create an immunosuppressive environment that allows the tumor to evade immune surveillance and grow. The YAP/TAZ pathway is activated in exhausted T-cells, and this activation may contribute to the development of T-cell exhaustion by promoting the expression of inhibitory receptors such as PD-1 that are associated with T-cell dysfunction [82,96,149,157,166,167,168,169,170,171,172,212,239,240]. Abbreviations: C-C motif chemokine ligand 2, CCL2; chemokine (C-X-C motif) ligand 1, CXCL1; chemokine (C-X-C motif) ligand 2, CXCL2; Granulocyte-macrophage colony-stimulating factor, GM-CSF; interleukin-6, IL6; myeloid-derived suppressor cells, MDSCs; tumor-associated macrophages, TAMs; transcriptional co-activator with PDZ-binding motif, TAZ; Yes-associated protein, YAP.

**Figure 6 cancers-15-03468-f006:**
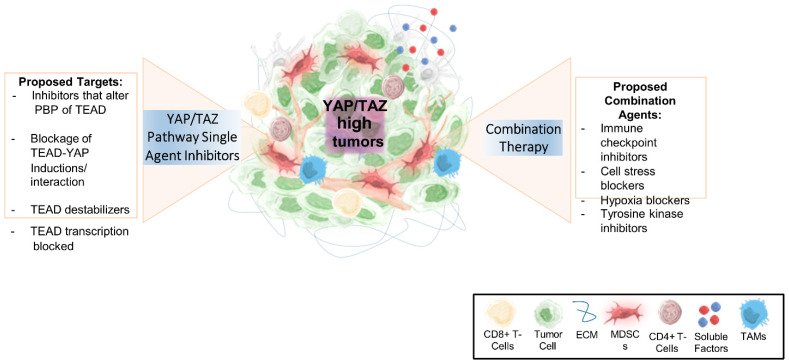
Therapeutic targeting of the YAP/TAZ/TEAD as a single agent or a combination approach. Although therapeutically targeting the YAP/TAZ pathway to treat cancer is currently in infancy, there is growing interest in targeting this pathway to Improve cancer treatment outcomes. Proposed targets include single-agent inhibition of critical components of the YAP/TAZ pathway or a combination approach that could include immune checkpoint inhibitors coupled with blocking regulators of the YAP/TAZ pathway (e.g., cell stress, hypoxia) [37,168,179]. Abbreviations: TEAD; Transcriptional Co-Activator with PDZ-Binding MOTIF; WW-Domain-Containing Transcription Regulator 1, WWTR1 (Also Known as TAZ); YES-Associated Protein, YAP.

**Figure 7 cancers-15-03468-f007:**
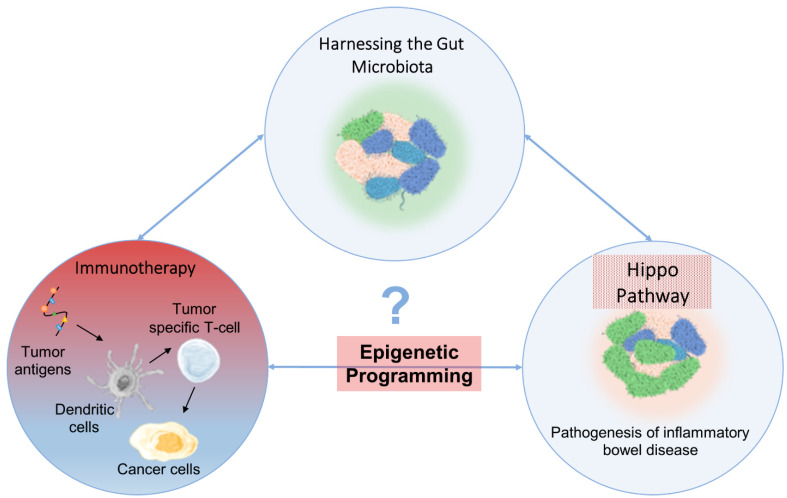
Hypothetical convergence of the gut microbiota and the Hippo pathway to enhance the effectiveness of immunotherapy. Although a clear linkage between the Hippo pathway, gut microbiota, and immunotherapy has not been established, there may be potential areas of convergence that can be leveraged to provide multi-modal effectiveness of immunotherapy in treating cancer [252,253,254,255].

## Data Availability

The data are contained within the article.

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
