# Peer review of "The Hippo Pathway Effectors YAP/TAZ-TEAD Oncoproteins as Emerging Therapeutic Targets in the Tumor Microenvironment"

_cancers, 2023, doi:10.3390/cancers15133468_

Round 1
Reviewer 1 Report
SWH and S.W.H should be consistent.
The author arbitrarily divides the research on the Hippo pathway into three decades. The diagram is informative, but I think it would be better to separate by milestone research.
A lot of content overlaps (e.g. "the upstream regulators" overlaps with "core Hippo kinase cassette" and "regulation by Rho GTPases"). The article needs to be better organized. One suggestion is to use one section to discuss the core kinase cascade, one section to discuss external stimuli (including mechanical stimuli, nutrients, etc.), and one section to discuss cross-talk between different signaling pathways.
Need to update YAP phosphorylation. In addition to S127, other phosphorylation sites with significant biological function have been identified.
Cell density should be included in the mechanical stimuli.
Much of the content in the "germline knockouts" section is actually YAP knock-in. "Transgenic" might be more accurate.
Most of the knock-ins or knock-outs should be tissue specific. Since organs contain multiple cell types, it is necessary to include the tissue specificity of the gene manipulation in this section (e.g., cardiomyocyte-specific YAP overexpression or global YAP overexpression).
"Early Embryonic Development" and "YAP/TAZ and Stem Cells" do not seem to belong in the "Germ Line Knockouts" section.
some of the sentences need improvement
Author Response
REVIEWER #1:
Comments and Suggestions for Authors
Firstly, we appreciate this reviewers’ critical inputs and suggestions. We have prepared this revised version that incorporates all the suggested changes by this reviewer. As you will see, we have adopted several changes that reorganize the review and improved the overall readability. As such, we hope that our improved version will be acceptable to this reviewer.
Comment 1-1: SWH and S.W.H should be consistent.
Response: Thank you. We have addressed this and confirmed consistency throughout the manuscript. Please review the following lines:
- Lines 43-47:
“YAP (Yes-associated protein) and TAZ (transcriptional co-activator with PDZ-binding motif) are components
of a more prominent pathway called the “Salvador-Warts-Hippo” (SWH) pathway (5).”
- Lines 62-63:
“The core components of SWH include the Hippo kinase bound to the adaptor protein Salvador, which activates the LATS/Warts kinase (5).”
------------------------------------------------
Comment 1-2: The author arbitrarily divides the research on the Hippo pathway into three decades. The diagram is informative, but I think it would be better to separate by milestone research.
Response: We appreciate this reviewer’s suggestion here. Accordingly, the figure 1 has been updated based on three key milestones. Please review the following line:
- Line 73:
“Figure 1: Significant Milestones in Hippo Pathway Discoveries from 1994 to 2022.” We have renamed each phase of discovery based on research milestones.
Comment 1-3: A lot of content overlaps (e.g. "the upstream regulators" overlaps with "core Hippo kinase cassette" and "regulation by Rho GTPases"). The article needs to be better organized. One suggestion is to use one section to discuss the core kinase cascade, one section to discuss external stimuli (including mechanical stimuli, nutrients, etc.), and one section to discuss cross-talk between different signaling pathways.
Response: Thank you for your comment. We have edited these sections and added subsections as suggested above. In this revised version, an initial section has been included that discusses the core kinase named as ‘The Core Hippo Kinase Cassette in Mammals’ on line 86. Following this, we have another section to assemble all the external stimuli affecting the Hippo pathway. This section is renamed as ‘Impact of extrinsic stimuli and mechanotransduction on YAP/TAZ’ on line 149. We have removed the crosstalk between the signaling pathways to make it more focused and concise.
Comment 1-4: Need to update YAP phosphorylation. In addition to S127, other phosphorylation sites with
significant biological function has been identified.
Response: Thank you for your feedback. We have accordingly revised the section ‘YAP/TAZ Nucleocytoplasmic Shuttling’ on line 300. The new information on novel phosphorylation sites of YAP/TAZ have been included here (line 310) as follows:
‘The consensus HXRXXS motif in YAP is phosphorylated by LATS1/2 in response to upstream inputs is conserved at all five residues including sites (S61, S109, S127, S164, and S381) (89), corroborative to the widespread observations that YAP5SA is strictly nuclear, overrides upstream regulation resulting in increased tumorigenesis. When YAP is mutated at the LATS 1/2 phosphorylation site (S127), inhibition of PDGF receptors causes YAP to move from the nucleus (active) to the cytosol (inactive). During the initial decade of Hippo pathway research, Serine 127 (S89 in TAZ) that affects nucleo-cytoplasmic localization and 381 (affects YAP degradation by sequential recognition by CK1 kinase and the E3 ligase SCFβ-TRCP) were identified as key phosphorylation sites that suppress the oncogenic functions associated with YAP (89). Since then, several additional serine and tyrosine phosphorylation sites have been identified that affect YAP functions. For instance, Nemo‐like kinase (NLK) have been shown to phosphorylate YAP at Ser128, which blocks the interaction with 14‐3‐3 and increases its nuclear localization (90). These emerging data is indicative of mutual exclusivity amongst Ser127 and Ser128 residues of YAP. Tyrosine phosphorylation, specifically, by SRC family kinases (SFK) targeting YAP at Y357, drives this shuttling. The primary function of SFK-mediated phosphorylation at Y357 is to retain YAP in the nucleus (49). Dasatinib inhibits SFK, causing YAP to migrate to the cytosol and decreasing YAP target genes. LCK, an SFK member, plays a key role in regulating YAPY357 phosphorylation, according to siRNA knockdown experiments, suggesting a novel function for LCK. This tyrosine phosphorylation method differs from serine phosphorylation, implying separate signals for YAP localization—and is also consistent with the notion that Dasatinib mediated inhibition of SFK impaired tumor growth and metastasis (91). In addition, c-ABL (Abelson murine leukemia viral oncogene) also phosphorylated YAP at tyrosine residue 357 to suppress its oncogenic functions and stabilize YAP-p73 interactions that induced cell apoptosis (92). Consistent with these observations, Src has also been shown to inhibit LATS by tyrosine phosphorylation that exerted enhanced YAP activity and also linked to intestinal inflammation (93, 94).
Comment 1-5: Cell density should be included in the mechanical stimuli.
Response: We have included the regulation of cell density shortly after mechanotransduction and provided a more detailed discussion.
The respective lines are 256-283. Besides including these section together as suggested by the reviewer, we have additionally named the section title for easy reference for our readers.
We also discussed mechanical stimuli and cell density in section titled “Reagents That Modulate the Subcellular Localization of YAP1/TAZ” on line 782.
Comment 1-6: Much of the content in the "germline knockouts" section is actually YAP knock-in. "Transgenic" might be more accurate.
Respond: Thank you for your comment. We have revised this title to reflect this feedback beginning on line 436. Please review the following line:
- Line 436:
“Transgenic and Germ Line Knockout Models”
Here we have categorized the genetic manipulation of YAP/TAZ based on tissue-specific functions.
------------------------------------------------
Comment 1-7: Most of the knock-ins or knock-outs should be tissue specific. Since organs contain multiple cell types, it is necessary to include the tissue specificity of the gene manipulation in this section (e.g., cardiomyocyte-specific YAP overexpression or global YAP overexpression).
Response: Thank you for this comment. We have gone through this paper and specified tissues of origin where the information was available. Please review the following line:
- Line 572-576:
Moreover, cancer cells use YAP/TAZ signalling to evade current therapeutics. In the liver of mice, the
overexpression of YAP has been demonstrated to block TNF-α induced cell death. At the same time, in breast
cancer, the knockdown of TAZ in MCF10A mammary cells improves susceptibility to taxol and other
chemotherapeutic drugs (168). This significant finding demonstrates how YAP1 can aid tumor cell escape even
with targeted chemotherapy.
------------------------------------------------
Comment 1-8: Early Embryonic Development" and "YAP/TAZ and Stem Cells" do not seem to belong in the "Germ Line Knockouts" section.
Response: Thank you for your comment. These two sections have been moved into a separate category. Please see the following lines:
- Lines 515-528:
“YAP/TAZ and Early Embryonic Development’’
Both YAP and TAZ play fundamental roles in embryonic development. Mutations that lead to a double null of
both proteins prevent embryos from surviving implantation (152). It is believed that YAP/TAZ has a critical
role in the homeostasis of the body axis, neural morphogenesis, and yolk sac vasculature; therefore, in mutants
of the proteins, abnormal and highly defective phenotypic changes are seen (153). For example, YAP/TAZ is
involved in cardiomyocytes' proliferative and regenerative properties (137). It has been reported that with YAP
overexpression, an increase in the overall size of the heart is noted. Moreover, a complete deletion of YAP was
shown to lead to thinning of ventricular walls to the extent that it led to lethal heart failure. By itself, TAZ
inactivation has no physiological symptoms; however, when combined with YAP inactivation, there is an
increase in the symptoms listed above (33). Consistent with the observation that inactivation of the Hippo
pathway reverses cardiac failure, the proteoglycan Agrin has been shown to promote cardiac regeneration via
promoting cardiomyocyte proliferation through Dystroglycan-Erk-YAP signaling (138, 139). The small
percentage of defective embryos that survive to term often die shortly later from organ dysfunctions listed above
(i.e., polycystic kidney disease) (150, 151).”
“YAP/TAZ and Stem Cells’’ on lines 530-545:
The proliferation of stem and progenitor cells is required for tissue homeostasis; therefore, they must be tightly
regulated. Countless studies have discovered that YAP and TAZ are needed for the tissue regeneration role of
stem cells (132, 146, 154). In mice tissue renewal, YAP and TAZ are required to maintain both the proliferation
and inhibition of differentiation of transit-amplifying (TA) cells, a population of cells in which there is an abundance in the expression of both molecules (78). Deleting YAP and TAZ can prevent both skeletal and
epithelial stem cell proliferation. At the same time, the expression of YAP is required for tissue homeostasis in
the liver, intestine, and mammary glands (78). The pathway identified in this process involves the activation of
the ITGA3-FAK-CDC42 signaling axis, in which Integrin α3 is inducted in TA cells leading to the
phosphorylation YAP-S397 which in turn activates the expression of Rheb and potentiates mTOR signalling
(78). The role of YAP/TAZ in the maintenance of stem cells is similar in human embryonic stem cells (hESCs).
Signals from TGFβ converge the SMAD pathway with the Hippo pathway to maintain pluripotency in hESCs
by suppressing the expression of differentiation markers (155). In Lgr5+ intestinal stem cells (ISCs), YAP
suppresses Wnt signaling, reprograming the cells to induce regenerative properties through EGFR signalling in
line with cancer initiation (156). Altogether, these findings demonstrate the regulatory role YAP/TAZ has in the
proliferation of stem cells and tissue homeostasis.”

Reviewer 2 Report
The author presents a comprehensive overview of the therapeutic potential of targeting the YAP/TAZ-TEAD oncoproteins in the tumor microenvironment. The authors provide a thorough comprehension of the YAP/TAZ oncogenic pathway and propose the development of inhibitors that interfere with YAP/TAZ signaling, with an emphasis on immune-microenvironmental regulations. In addition, the role of the YAP/TAZ pathway in modulating adaptive immune system cells and the expression of cytokines and chemokines that influence immune cell trafficking and recruitment to the tumor microenvironment is highlighted.
Overall, the timeliness and veracity of this manuscript are acceptable for a review.
I do recommend that the authors make the preponderance of suggested modifications to the paper. Here are some important considerations:
Insufficient detail is provided in the review of regulatory factors upstream of the hippo cascade in mammals. For instance, the analysis of regulatory factors upstream of the hippo cascade in mammals is insufficiently comprehensive. Tao Li et al. (PMC8288455) found that the YAP phosphorylation inhibitor DLG2 difference in the differentially expressed genes between two distinct environment-adapted cell lines was associated with cell survival. This factor's cascade effect on the YAP is also related to cell density. This also relates to the "Regulation by Cell Density and Epithelial Architecture" section of this review. As there is a strong correlation between cell density and the hippo cascade, the pertinent material should be included in this review.
Author Response
REVIEWER #2:
Comments and Suggestions for Authors
The author presents a comprehensive overview of the therapeutic potential of targeting the YAP/TAZ-TEAD oncoproteins in the tumor microenvironment. The authors provide a thorough comprehension of the YAP/TAZ oncogenic pathway and propose the development of inhibitors that interfere with YAP/TAZ signaling, with an emphasis on immune-microenvironmental regulations. In addition, the role of the YAP/TAZ pathway in modulating adaptive immune system cells and the expression of cytokines and chemokines that influence immune cell trafficking and recruitment to the tumor microenvironment is highlighted. Overall, the timeliness and veracity of this manuscript are acceptable for a review. I do recommend that the authors make the preponderance of suggested modifications to the paper. Here are some important considerations:
Response: We appreciate this reviewer’s overall insights and support for our review article. According to the comments provided, we have further revised this version to improve the robustness of the article.
Comment 2-1: Insufficient detail is provided in the review of regulatory factors upstream of the hippo cascade in mammals. For instance, the analysis of regulatory factors upstream of the hippo cascade in mammals is insufficiently comprehensive. Tao Li et al. (PMC8288455) found that the YAP phosphorylation inhibitor DLG2 difference in the differentially expressed genes between two distinct environment-adapted cell lines was associated with cell survival. This factor's cascade effect on the YAP is also related to cell density. This also relates to the "Regulation by Cell Density and Epithelial Architecture" section of this review. As there is a strong correlation between cell density and the hippo cascade, the pertinent material should be included in this review.
Response: Thank you for your comments. We have included some further content on this section. Please review the following lines:
- Lines 123-140: included in the section ‘Upstream regulators of the Hippo Cascade in Mammals’
Cell-cell adhesion molecules, such as E-cadherin and α-catenin, have been implicated in the regulation of the Hippo pathway. Disruption of the E-cadherin and α-catenin complexes has also been shown to interfere with the phosphorylation of YAP, decreasing it, which in turn increases the amount of YAP found in the nucleus (35). These molecules contribute to the establishment of cell polarity and formation of adherens junctions. Adherens junctions, in turn, regulate the Hippo pathway by influencing the localization and activity of components such as Merlin (also known as NF2), which acts upstream of MST1/2 and LATS1/2 to inhibit YAP activity. In addition, members of the membrane-associated guanylate kinase (MAGUK) protein discs large homolog DLG family connect cell polarity and Hippo signaling through PAR-1 linkage with MST1/2 (55). While the phenotypic states of cancer cells are largely dynamic, often switching between the classical ‘r’ states that are proliferative and responsive to cell density and the ‘K selection’ state that have limited growth potential (37). Interestingly, Li et al., found that the YAP phosphorylation inhibitor DLG2 differentially impacts in the YAP/TAZ transcriptome between two distinct environment-adapted cell lines was associated with cell survival (56). The study found that the presence of the YAP phosphorylation inhibitor DLG2 was associated with cell survival in differentially expressed genes between two environment-adapted cell lines. DLG2's inhibitory effect on YAP phosphorylation protected cells and contributed to their adaptive response to diverse environments. These findings highlight DLG2's role as an upstream regulatory factor in the Hippo pathway, influencing tumor behavior and functional diversity.
All other details regarding the discussion on the regulation by cell density are updated. We have also included the role of MAGUK proteins in the differential regulation of YAP/TAZ based on cell states here.

Reviewer 3 Report
This review by Mokhtari et al provides recent insights on the role of YAP/TAZ as effectors of the Hippo pathway in healthy versus the cancer microenvironment. The authors also discussed the therapeutic strategies intervening in each of the Hippo signaling modules for improved cancer treatments.
Line 88-107: Could the author change the subtitle in line 88 since only one sentence mentioned cancer? Indeed, the cancer-related content could be merged with the later section where cancer is the primary focus, section starting from line 475.
Line 259: The reviewer suggests the authors focus on the Hippo pathway first before discussing the crosstalk of Hippo with other pathways.
Line 278: It seems that YAP/TAZ shuttling (line 89) and this section could be combined together.
There are two figures labeled with Figure 2. Although there is Figure 2 A and Figure 2 B in the captions, it is kind of confusing why these two figures are not merged together since the author would like to use Figure 2 for both of them.
Line 333: Would it be better if the authors move this section under the section beginning from line 373?
Figure 3: RAP2 is included in the figure but not discussed. The reviewer suggests including this significant paper on RAP2 Mediates Mechano-responses of Hippo Pathway (PMCID: PMC6128698)
Line 600&631: The subtitle format is not consistent with other subtitles.
Author Response
REVIEWER #3:
Comments and Suggestions for Authors
This review by Mokhtari et al provides recent insights on the role of YAP/TAZ as effectors of the Hippo pathway in healthy versus the cancer microenvironment. The authors also discussed the therapeutic strategies intervening in each of the Hippo signaling modules for improved cancer treatments.
We sincerely thank this reviewer for his/her constructive comments. We hope that our revised version addresses all the changes as requested by the reviewer.
Comment 3-1: Line 88-107: Could the author change the subtitle in line 88 since only one sentence mentioned cancer? Indeed, the cancer-related content could be merged with the later section where cancer is the primary focus, section starting from line 475.
Response: Thank you for your comments. We have revised the section accordingly. The regulation of YAP/TAZ in normal versus tumors are discussed later in line 397. In addition, we have moved the section of YAP/TAZ nucleocytoplasmic shuttling on line 300 to make it consistent with the subsequent discussions on the nuclear role of YAP/TAZ complexes.
Comment 3-2: The reviewer suggests the authors focus on the Hippo pathway first before discussing the
crosstalk of Hippo with other pathways.
Response: Thank you for your comments. We have reorganized our review article. It first discusses the The Core Hippo Kinase Cassette in Mammals on line 86, followed by ‘Upstream regulators of the Hippo Cascade in Mammals’: line 98, and followed by subsequent sections. Because there are diverse signaling pathways that crosstalk with YAP/TAZ, we have omitted this section as this warrants a separate review which is beyond the scope of the current version. We appreciate kind understanding from the reviewer.
Lines 373-375 in the previous version is now presented as part of the Figure 3 and section ‘Nuclear YAP/TAZ Complexes and transcriptional output’ beginning from line 335. Accordingly, we have also revised the respective figure legend as follows:
Figure 3 legend: ‘SWI/SNF and ARIDIA control DNA accessibility, while nuclear actin enhances transcriptional activity. These factors are critical in YAP/TAZ-mediated transcriptional regulation and oncogenic transformation.
-------------------------------------------------------
Comment 3-3: It seems that YAP/TAZ shuttling (line 89) and this section could be combined together.
Response: Thank you for the insightful suggestion. Accordingly, the section from line 89 has been moved to line 300.
Comment 3-4: There are two figures labeled with Figure 2. Although there is Figure 2 A and Figure 2 B in the captions, it is kind of confusing why these two figures are not merged together since the author would like to use Figure 2 for both of them.
Response: Once again, thank you for your input. Given that the figures discuss two aspects of the Hippo pathway, we have re-numbered the figures (Figure 2 and Figure 3) to avoid confusion and allow us to draw special attention to the nuclear signaling.
Comment 3-5: Figure 3: RAP2 is included in the figure but not discussed. The reviewer suggests including this significant paper on RAP2 Mediates Mechano-responses of Hippo Pathway (PMCID: PMC6128698).
Response: Many thanks for pointing this out. This has been added to the figure 4 caption. Please review the following lines:
- Lines 422-423:
“At low ECM stiffness, active RAP2 promotes Rho GTPase activating protein 29 (ARHGAP29) and mitogen-activated protein kinase (MAP4K4/6/7) which in turn results in YAP/TAZ inactivation (126)”
We have also added the following in lines 251-254
‘Intracellular mechanotransduction is also mediated by RAP2, a Ras-related protein, which is activated by low stiffness by phosphatidic acid (PA) and inactivates YAP under low stiffness conditions (135). Mechanistically, the YAP inhibition under compliant ECM was achieved through through MAP4K4/6/7 and ARHGAP29.’
Comment 3-6: Would it be better if the authors move this section under the section beginning from line 373?
Response: Thank you for your comment. We have moved the portion on the heart to this section per your suggestion. Please review the following lines:
- Lines 520-528:
“For example, YAP/TAZ is involved in cardiomyocytes' proliferative and regenerative properties (138). It has been reported that with YAP overexpression, an increase in the overall size of the heart is noted. Moreover, a complete deletion of YAP was shown to lead to thinning of ventricular walls to the extent that it led to lethal heart failure. By itself, TAZ inactivation has no physiological symptoms; however, when combined with YAP inactivation, there is an increase in the symptoms listed above (33). Consistent with the observation that inactivation of the Hippo pathway reverses cardiac failure, the proteoglycan Agrin has been shown to promote cardiac regeneration via promoting cardiomyocyte proliferation through Dystroglycan-Erk-YAP signaling (139, 140). The small percentage of defective embryos that survive to term often die shortly later from organ dysfunctions listed above (i.e., polycystic kidney disease) (151, 152).”
----------------------------------------------
Comment 3-7: The subtitle format is not consistent with other subtitles.
Response: Thank you for your comment. We Revised the subtitles to match other subtitles in this section. Please review the following lines:
- Line 782:
“Reagents That Modulate the Subcellular Localization of YAP1/TAZ”
In addition, we have revised the section titles of several portions of the manuscript. Please see the following:
‘Role of YAP/TAZ in normal versus tumor-vasculature’ on line 654.

Reviewer 4 Report
Recently, a large number of reviews have documented the various and distinct oncogenic and tumor suppressive modules which constitute the HIPPO pathway. Among the various signaling in which the HIPPO pathway has been studied is that of the tumor microenvironment (TME). The knowledge of the mechanisms through which the tumor is able to support its growth within a tissue and undertake an invasion in it is precious for implementing new therapeutic strategies.
The present manuscript describes the recent literature on the involvement of the HIPPO pathway and its transcriptional effectors in influencing TME.
Very recently diverse reviews have been published on the same topic describing the same aspects and therapeutic perspectives. In the last year there hasn't been an explosion of research articles in the field or highly innovative therapies high enough to warrant this review. This concept is also evident from the bibliography of the authors.
Overall, the review is well written and well structured, captivating in the graphic models, however it does not contain a large novelty of topics to be explored already proposed in the recent review bibliography on the same topic.
It is my opinion that it cannot be accepted on Cancers as there is no search for a new aspect to discuss in a field that risks becoming asphyxiated by reviews that are all the same.
- Piccolo S, Panciera T, Contessotto P, Cordenonsi M. YAP/TAZ as master regulators in cancer: modulation, function and therapeutic approaches. Nat Cancer. 2023;4(1):9-26. Epub 2022/12/24. doi: 10.1038/s43018-022-00473-z.
- Cunningham R, Hansen CG. The Hippo pathway in cancer: YAP/TAZ and TEAD as therapeutic targets in cancer. Clin Sci (Lond). 2022 Feb 11;136(3):197-222. doi: 10.1042/CS20201474.
- Ortega Á, Vera I, Diaz MP, Navarro C, Rojas M, Torres W, Parra H, Salazar J, De Sanctis JB, Bermúdez V. The YAP/TAZ Signaling Pathway in the Tumor Microenvironment and Carcinogenesis: Current Knowledge and Therapeutic Promises. Int J Mol Sci. 2021 Dec 31;23(1):430. doi: 10.3390/ijms23010430.
- Fu M, Hu Y, Lan T, Guan K-L, Luo T, Luo M. The Hippo signalling pathway and its implications in human health and diseases. Signal Transduction and Targeted Therapy. 2022;7(1):376. doi: 10.1038/s41392-022-01191-9.
- Yang D, Zhang N, Li M, Hong T, Meng W, Ouyang T. The Hippo Signaling Pathway: The Trader of Tumor Microenvironment. Front Oncol. 2021 Nov 11;11:772134. doi: 10.3389/fonc.2021.772134.
Author Response
REVIEWER #4:
Recently, a large number of reviews have documented the various and distinct oncogenic and tumor suppressive modules which constitute the HIPPO pathway. Among the various signaling in which the HIPPO pathway has been studied is that of the tumor microenvironment (TME). The knowledge of the mechanisms through which the tumor is able to support its growth within a tissue and undertake an invasion in it is precious for implementing new therapeutic strategies. The present manuscript describes the recent literature on the involvement of the HIPPO pathway and its transcriptional effectors in influencing TME. Very recently diverse reviews have been published on the same topic describing the same aspects and therapeutic perspectives. In the last year there hasn't been an explosion of research articles in the field or highly innovative therapies high enough to warrant this review. This concept is also evident from the bibliography of the authors. Overall, the review is well written and well structured, captivating in the graphic models, however it does not contain a large novelty of topics to be explored already proposed in the recent review bibliography on the same topic.
It is my opinion that it cannot be accepted on Cancers as there is no search for a new aspect to discuss in a field that risks becoming asphyxiated by reviews that are all the same.
- Piccolo S, Panciera T, Contessotto P, Cordenonsi M. YAP/TAZ as master regulators in cancer: modulation, function and therapeutic approaches. Nat Cancer. 2023;4(1):9-26. Epub 2022/12/24. doi: 10.1038/s43018-022-00473-z.
- Cunningham R, Hansen CG. The Hippo pathway in cancer: YAP/TAZ and TEAD as therapeutic targets in cancer. Clin Sci (Lond). 2022 Feb 11;136(3):197-222. doi: 10.1042/CS20201474. - Ortega Á, Vera I, Diaz MP, Navarro C, Rojas M, Torres W, Parra H, Salazar J, De Sanctis JB, Bermúdez V. The YAP/TAZ Signaling Pathway in the Tumor Microenvironment and Carcinogenesis: Current Knowledge and Therapeutic Promises. Int J Mol Sci. 2021 Dec 31;23(1):430. doi: 10.3390/ijms23010430.
- Fu M, Hu Y, Lan T, Guan K-L, Luo T, Luo M. The Hippo signalling pathway and its implications in human health and diseases. Signal Transduction and Targeted Therapy. 2022;7(1):376. doi: 10.1038/s41392-022-01191-9.
- Yang D, Zhang N, Li M, Hong T, Meng W, Ouyang T. The Hippo Signaling Pathway: The Trader of Tumor Microenvironment. Front Oncol. 2021 Nov 11;11:772134. doi: 10.3389/fonc.2021.772134.
Response: We would like to thank you for taking the time to review our manuscript on the involvement of the Hippo pathway and its transcriptional effectors in the tumour microenvironment. We appreciate your positive feedback regarding the quality of the writing, structure, and graphical models. We acknowledge your concern that there have been several recent reviews on a similar topic; however, we would like to emphasize that our recent work presents novel findings and perspectives that have not been covered extensively in existing review papers. Our review contributes highlights recent advancements in understanding the interplay between the Hippo pathway, the immune system, and immunotherapy within the tumour microenvironment. The distinct contributions of our review touch on three key domains that have not been collectively explored in the current body of literature. As such, this is the first comprehensive review (citing over 300 related works in the field of Hippo pathway and YAP/TAZ) that assembles all these unique following topics:
- Exploration of Combination Therapy and Tumour Microenvironment: We delve into the impact of combination therapy on the Hippo pathway, particularly in the context of hypoxia and the tumour microenvironment. We discuss how targeting the Hippo pathway in combination with other therapeutic modalities can affect its activation and influence tumour response to treatment. This comprehensive analysis sheds light on the interplay between the Hippo pathway and the complex tumour microenvironment, providing insights into potential strategies to overcome treatment resistance.
- Impact of the Hippo Pathway on immunotherapy and the tumour immune microenvironment (TIME): In recent years, emerging evidence has shown that the Hippo pathway plays a crucial role in modulating immune responses within the tumour microenvironment. The Hippo pathway components have been found to interact with immune cells and modulate their function. This crosstalk between the Hippo pathway and the immune system has profound implications for tumour immune evasion, antitumor immunity, and response to immunotherapy. Our review synthesizes and presents the most recent findings regarding the interaction of the Hippo pathway with various immune cell populations, including T cells, natural killer cells, macrophages, and dendritic cells, within the tumour microenvironment. We discuss how Hippo pathway dysregulation can impact immune cell infiltration, activation, and polarization, ultimately shaping the immunosuppressive or immunostimulatory nature of the TME. Furthermore, we explore the potential of targeting the Hippo pathway components as a strategy to enhance immunotherapy efficacy, including immune checkpoint blockade and adoptive T-cell therapy.
- Consideration of new emerging concepts merging epigenetics and the microbiome: We recognize the importance of epigenetic mechanisms and the microbiome in the regulation of the Hippo pathway. Our review addresses the influence of epigenetic modifications and microbial factors on Hippo pathway activation and their potential impact on tumour progression and therapeutic responses. This integration of epigenetic and microbiome perspectives provides a comprehensive understanding of the Hippo pathway regulation, highlighting new avenues for therapeutic interventions. The incorporation of recent research and our group’s extensive work in multifactorial strategies to improve immunotherapy outcomes allows us to present novel perspectives and insights into the complex interplay between the Hippo pathway and the immune system, emphasizing the need for further investigation in this area.
Once again, we sincerely appreciate your feedback and we hope that our work, despite the presence of other reviews on related topics, can still make a valuable contribution to the field and provide insights into the implications of the Hippo pathway on immunotherapy outcomes and connection to the gut microbiota.

Round 2
Reviewer 2 Report
The authors responded to all of my concerns carefully. I have no further questions.
Author Response
Many thanks for supporting our work
Reviewer 3 Report
The authors have successfully addressed my concerns. I have no further comments.
Author Response
We appreciate your insightful comments and thank you for helping us revise the manuscript
Reviewer 4 Report
The present revised version of the manuscript is much better structured than the first one, showing more effectively the most recent aspects of the HIPPO pathway findings, also with regard to possible targeted and immunomodulatory therapies. The authors have made a convincing revision and update effort well described in the rebuttal letter.
This version is acceptable for publication.
Author Response
Many thanks for your inputs and efforts in helping us to create a more robust revised article. Much appreciated.